# Physicians’ Trust in Relevant Institutions during the COVID-19 Pandemic: A Binary Logistic Model

**DOI:** 10.3390/healthcare11121736

**Published:** 2023-06-13

**Authors:** Tudor-Ștefan Rotaru, Aida Puia, Ștefan Cojocaru, Ovidiu Alexinschi, Cristina Gavrilovici, Liviu Oprea

**Affiliations:** 1Department of Bioethics, University of Medicine and Pharmacy “Gr. T. Popa” Iași, 700115 Iași, Romania; tudor.rotaru@umfiasi.ro (T.-Ș.R.); liviu.oprea@umfiasi.ro (L.O.); 2Department of Community Medicine, “Iuliu Hațieganu” University of Medicine and Pharmacy, 400347 Cluj-Napoca, Romania; 3Department of Sociology and Social Work, University “Alexandru Ioan Cuza” of Iași, 700506 Iași, Romania; contact@stefancojocaru.ro; 4Department IIIA, “Socola” Institute of Psychiatry, 700282 Iași, Romania; alexinschi@yahoo.com; 5Department of Mother and Child, University of Medicine and Pharmacy “Gr. T. Popa” Iași, 700115 Iași, Romania; cri.gavrilogici@umfiasi.ro

**Keywords:** responsibility, trust, willingness to work

## Abstract

Little research has been done on professionals’ perceptions of institutions and governments during epidemics. We aim to create a profile of physicians who feel they can raise public health issues with relevant institutions during a pandemic. A total of 1285 Romanian physicians completed an online survey as part of a larger study. We used binary logistic regression to profile physicians who felt they were able to raise public health issues with relevant institutions. Five predictors could differentiate between respondents who tended to agree with the trust statement and those who tended to disagree: feeling safe at work during the pandemic, considering the financial incentive worth the risk, receiving training on the use of protective equipment, having the same values as colleagues, and enjoying work as much as before the pandemic. Physicians who trusted the system to raise public health issues with the appropriate institutions were more likely to feel that they shared the same values as their colleagues, to say they were trained to use protective equipment during the pandemic, to feel that they were safe at work during the pandemic, to enjoy their work as much as before the pandemic, and to feel that the financial bonus justified the risk.

## 1. Introduction

The COVID-19 pandemic demonstrated the importance of communicating effective public health interventions to both the public and to healthcare workers. Next to effective and fast communication, a critical factor in the success of a public health intervention is trust. For instance, in a Korean study, trust in government information was identified as a primary determinant of willingness to use a mobile application for public health risk communication [1]. Citizens’ trust in public health policies is of paramount importance to their willingness to comply with measures, especially if those measures are more difficult to implement and respect. A study conducted in Liberia found that respondents who expressed low trust in government were much less likely to take precautions against the Ebola virus in their homes, less likely to comply with government-mandated social distancing mechanisms designed to limit the spread of the virus, and much less likely to support potentially controversial control measures such as “safe burial” of infected bodies [2]. It has been suggested that governments can use information intermediaries to gain public trust. These intermediaries can be experts meant to enhance the credibility of information presented to the public. Their presence in the public forum should increase public confidence and motivate citizens to comply with health policies [3]. It follows that professionals have a unique role when it comes to mediating public trust in government. This is especially important during public health emergencies, when rapid communication is needed. In turn, professionals must have confidence in the health policies they advocate. Assessing professionals’ trust in national policies is essential to the success of public health interventions [4].

During a pandemic, public health officials must make decisions quickly. For instance, during the COVID-19 pandemic, interventions had to be developed and evaluated with less information and less time than usual. However, health professionals are generally trained in evidence-based medicine, which takes time to test, validate, and translate into clinical guidelines. Therefore, despite the urgency of the public health situations faced by clinicians, evidence-based science has played a key role in informing decisions in the development and evaluation of COVID-19 public health interventions. In addition, evidence-based science has played a key role in credibility, viewed in all its aspects. Therefore, another important aspect is access to sufficient sources that are credible to both professionals and the public. Not surprisingly, identifying the sources used can enhance the credibility of health policy-making organizations. When sources are identified, decision-making can be made more transparent to both the public and health professionals [4].

There is already a substantial body of research on public opinion about public health interventions. However, little research has been done on the perception of professionals during epidemics, although the latter is just as important. We argue that in such a situation professionals play a crucial role in both decision-making and public trust. On the one hand, the success of public health interventions during an epidemic is linked to public trust in health professionals. On the other hand, public trust depends on the public’s belief that professionals are genuinely involved in the decision-making process [5,6]. Unfortunately, we know that public trust in decision-makers has declined over recent years. For instance, Matthew Bennett argued that the success of the public health response to the COVID-19 pandemic depended on public trust in experts. According to this author’s argument, when public policy claims to follow science, citizens are asked not only to believe what experts tell them, but to follow experts’ recommendations. Bennett argues that this requires a more sophisticated form of trust, which he calls recommendation trust. The conditions for recommendation trust are more demanding than the conditions for epistemic trust. His conclusion is that many of the measures that have been proposed to cultivate trust in experts do not provide the public with good reasons to trust in expert-led policy [5]. Not surprisingly, the lack of trust in science itself may have increased public distrust in evidence-based health policy. It is therefore important to understand physicians’ perceptions of trust in policy makers. A key element in this endeavor is to examine physicians’ trust in government agencies [4].

A study conducted in Israel on 112 public health workers assessed trust using 5-point Likert scales in May 2020, during the COVID-19 outbreak. One interesting finding was that public health physicians had lower trust than researchers and other health professionals. The same study found that professionals with low levels of trust were less likely to use government tools to track infected patients. These same doctors had lower levels of trust in the Ministry of Health. Most health workers in the survey rated their involvement in decision-making as low or non-existent. At the same time, they reported lower levels of trust in politics than those with high levels of involvement [4].

Another Israeli study conducted during the same period examined the factors influencing physicians’ decision-making and preventive behaviour during the COVID-19 pandemic. The study examined the responses of 187 Israeli physicians in April and May 2020 based on a complex questionnaire that included perceived risk during the pandemic, emotions, trust in the health care system, and physician compliance with preventive measures. The participants in the study were asked to indicate the level of negative and positive emotions they had experienced in the past week on a scale ranging from “not at all” to “extremely”. The negative emotions included fear, anger, anxiety, stress, nervousness, bad mood, blame, and frustration, while the positive emotions included enthusiasm, relaxation, strength, “sense of mission”, pride, and activism. The researchers found that higher levels of trust in public institutions were associated with higher levels of compliance with Ministry of Health guidelines, more positive emotions, and more cautious decision making among physicians [7].

An older Japanese study [8] on the 2009 influenza pandemic is particularly relevant to this paper. This cross-sectional survey of willingness and reluctance to work during the H1N1 pandemic collected responses from 3635 employees at three core hospitals in the city of Kobe, Japan. The most influential factor that motivated people to work was the feeling of being protected by their country, local government, and hospitals. Conversely, workers who were more reluctant to work were concerned about becoming infected, compensation if they became infected, and feeling isolated. The authors concluded that professionals’ trust in public organizations significantly influenced their willingness to work during a pandemic. These findings show that physical protection against infection is important both in its own right and in relation to trust. Lessons from the H1N1 pandemic may be valuable for the COVID-19 pandemic [9].

Another study examined physicians’ trust in government agencies and scientists as a variable influencing their effectiveness in communicating health care policy to patients. A total of 625 primary care physicians completed an online survey that, among other things, examined trust in media outlets as opposed to trust in the government. The study challenged the assumption that doctors, by virtue of their expertise, provide accurate and up-to-date information to their patients. Interestingly, the researchers found that physicians are subject to the same biases that influence public opinion. Doctors’ trust in the media influenced their concern that a family member might get sick. In addition, it influenced their perceptions of the severity of the pandemic and their trust in government agencies and scientists [10].

It is interesting to note that researchers studying the 2014–2015 Ebola virus in Liberia have suggested a possible vicious cycle between distrust, noncompliance, hardship, and further distrust at the public level. The research team conducted a study of approximately 1600 participants from the general public between 6 December 2014 and 7 January 2015 in Monrovia, Liberia. Questions were asked about the perceived responsiveness and trustworthiness of public institutions’ (e.g., the Liberian Ministry of Health) compliance with control measures (e.g., keeping a bucket of chlorinated water at home), and self-reported support for policies (e.g., the night-time curfew). Several of the findings of this study support the idea that respondents who refused to comply with public health policies may have done so because they did not trust the ability or integrity of government institutions to recommend precautions and implement policies to slow the spread of the Ebola virus.. Respondents who experienced hardship, such as losing a job or going without health care, expressed much less trust in government than those who did not. While the result is only correlational, it raises the possibility of a vicious cycle. Those who experienced hardship were less likely to trust the government, and consequently less likely to comply with Ebola control measures. These people may have put themselves at greater risk of infection. Being infected meant a higher risk of experiencing further hardship [2]. The vicious circle model proposed by Blair and colleagues may apply to health care professionals involved in pandemics as well. Struggling physicians may have less confidence in government policies. To them, certain policies may seem meaningless. If they become infected by not following these measures, the risk of further difficulties increases. Our study included items related to the number of children in care, self-assessment of socioeconomic status, housing situation, and the number of infections or hospitalizations the health worker had suffered as a result of COVID-19. If our data were to support Blair et al.’s vicious cycle model, predictors might include the difficulties identified by such things.

The literature shows that trust is important for the success of public health interventions during a pandemic. Using different methodologies, the literature identifies different factors that influence trust. These papers focus on public trust in institutions, not on physicians’ trust in institutions in the context of pandemics. We were able to identify two problems. First, little has been written about physicians’ trust in institutions in pandemic contexts. Second, the list of factors already suggested to influence this trust is short. The present paper contributes to both issues by identifying additional factors and by focusing specifically on physicians’ trust in government institutions.

The literature shows that greater trust in public institutions on the part of both the public and health care professionals increases the likelihood of effective health care policy implementation. This is even more true during epidemics. Studies to date have measured various factors that may contribute to this trust: the use of experts, the promotion of credible sources, the public’s belief that professionals are genuinely involved in the decision-making process, and professionals’ own belief that they are involved in the decision-making process. These factors have been shown to increased trust in evidence-based science by promoting a sense that health worker are protected and encouraging the avoidance of biases promoted by the media. Moreover, these factors foster trust in the integrity and ability of institutions to implement effective policies. However, we are far from developing a coherent theoretical model that takes all these factors into account. The present paper aims to expand the list of factors that may influence health care professionals’ trust in public institutions. Professionals’ trust in government agencies has been less studied than public trust. This paper contributes to the literature gap on factors influencing health care professionals’ trust in the public institutions implementing health policies. The more we know about the factors that influence health care professionals’ trust, the more these factors can be influenced in future epidemics. Because the context of an epidemic requires rapid intervention, knowing such factors in advance can pave the way for a more rapid response. Changing the factors that influence physicians’ trust in government institutions can be accomplished early; if all these trust interventions prove effective, physicians will be more likely to respond favorably to public health interventions when the need arises.

The aim of this study was to profile physicians willing to raise public health issues with relevant institutions during the COVID-19 pandemic. We used an online questionnaire consisting of questions on socio-demographic and socio-economic status, self-reported diseases, medical specialties, health status, place of work, and involvement in pandemic-related activities. Likert-type scales were used to measure constructs such as self-efficacy, duty, willingness to work, and institutional and governmental reciprocity. A total of 1285 Romanian physicians participated in the study. Binary logistic regression was used to determine the best predictors for the two categories of responses (i.e., higher and lower trust in relevant institutions). A total of 74 items were entered into the regression equation, and significant differences were found between the two categories. This study can provide insight into the factors that influence a physician’s willingness to address public health issues during a pandemic.

## 2. Materials and Methods

In this study, we used binary logistic regression to create a profile of physicians who felt they could raise public health issues with relevant institutions. The online questionnaire began with an informed consent form that the participant had to agree to. The consent form provided details about the purpose of the study, the research team, and the methodology used to collect the data, and included the following statement: “None of the information you provide will be linked to you as an individual. […] There is no risk to the survey other than you may become more aware of the impact the COVID-19 pandemic may have had on your well-being as well as your professional experience during this time”. Regarding privacy, the section included the following assurance: “By agreeing to participate in this research, you are not jeopardizing any legal rights. All information from your participation in this research will be collected and stored in accordance with the General Data Protection Regulation (GDPR).” The end of the consent section contained the following statement: “By pressing the CONTINUE button, I agree to participate in this study.

The questionnaire, administered online, was structured in the following parts. From question 1 to question 18, socio-demographic items (e.g., gender, age, region in Romania where the doctor worked, marital status) and items related to socio-economic status (number of people with whom the doctor lived, number of elderly people in the household, number of young children, number of rooms in the household, self-assessment of economic status on a scale from 1 to 10, whether the doctor is currently employed, etc.) were included. Question 19 targeted self-reporting of the various diseases the doctor suffered from. Because in the online questionnaire the respondents had to select from a list of multiple options, in the database we unpacked each option into a separate dichotomous variable (e.g., for question 19t: “Does the medical professional self-report bronchial asthma, including allergic asthma”, respondents could answer YES = 1 or NO = 0). Without this decomposition into dichotomous variables, it would have been impossible to process the information resulting from question 19. Further, question 20 to 25 asked about the medical professional’s specialty and self-assessment of health status in the last two weeks respective to the two weeks before the pandemic. Question 26 targeted the specific place of work of the health care professional (e.g., maternity, intensive care, day hospital, etc.). Because two or more of these options could be present at the same time, we split each option into a separate dichotomous variable in a similar way to question 19. In questions 27 to 36, the healthcare professional was asked whether he/she worked in a quarantined locality, the average number of hours worked per week during the pandemic, and whether he/she was hospitalised as a result of a COVID infection. Question 37 referred to the types of activities in which the healthcare professional was involved during the pandemic (e.g., collecting biological samples from suspected COVID-19 patients, performing aerosol-generating procedures, examining a patient with respiratory symptoms, etc.). This question was broken down into several dichotomous variables, as the same physician might have performed one or more of these activities. Question 99 asked for a ranking from 1 (most important) to 3 (least important) of factors that negatively influenced medical practice (e.g., infection control procedures, discharge of non-emergency patients from hospital, etc.). Questions 129 and 130 asked for a ranking of the main sources of information (e.g., direct supervisors, other colleagues in the community, etc.) and main supportive factors relied on (e.g., colleagues at work, professional organizations, etc.) when the doctor faced a professional problem. Questions 99, 129, and 130 were in turn broken down into separate variables for each option, with the number corresponding to the hierarchy (1, 2 or 3) being recorded. The remaining questions 38–144 were answered on 6-point Likert-type scales, and referred to various constructs in the literature, i.e., self-efficacy, duty, willingness to work, institutional reciprocity, and government reciprocity. Due to the lengthy process of constructing the questionnaire, these items were not grouped in the database according to the construct they were part of. However, each item was labelled as such in the database (e.g., Question 49: “I was aware of my role during the pandemic”—Self-efficacy; Question 43: “A healthcare professional who does not have children should work on the front lines during a pandemic” (reversed item)—Duty). After data collection, due to multiple situations that did not apply to one or more of the physicians, the research team did not aggregat the items in the questionnaires and did not calculate the internal consistency coefficient and total per scale. All the data were processed in IBM SPSS Statistics 20.0.

A total of 1285 Romanian doctors completed an online survey between July and August 2020 as part of a larger study on responsibility, medical ethics, willingness to work, and self-efficacy during the first wave of the COVID-19 pandemic [11,12,13,14,15]. The sample was nationally relevant both demographically and in terms of the distribution of physicians among the different regions of Romania. Of the total number of respondents, 982 were female and 302 were male, with one participant not specifying their gender. The gender distribution in the sample was relevant to the gender distribution of physicians nationwide. The mean age of the sample was 48.21 years, with a minimum of 25 years and a maximum of 86 years. The participants belonged to all known specialties in the field of medicine.

The total number of respondents can be considered a convenience sample, although it is intended to be as close as possible to national representativeness. The questionnaires were distributed via the internet through the local offices of professional associations with the intention of covering as much of the country as possible. The exact response rate is not known. It is difficult to calculate the proportion of respondents among those invited to respond. This is due to the fact that the questionnaire was distributed through professional networks and different branches of professional associations. However, the composition of the sample by region of the country and the urban–rural distribution can be presented. Table 1 shows the distribution of the respondents to the questionnaire categorized by the historical regions of the country (Transylvania, Banat, Crișana, etc.) and by type of community (village, small town, medium town, large town). The most important selection bias was the self-selection of the participants. The fact that the questionnaire was completed anonymously was an attempt to counter this effect; however, self-selection bias could not be completely eliminated. It is possible that extremely busy and extremely dissatisfied doctors may have responded in a lower proportion. These biases are a limitation of the study.

In addition, the fact that the questionnaire was distributed through professional associations and networks may be another source of bias. It is possible that physicians who were better connected to their colleagues in professional associations may have participated more. Physicians who use the internet less, those who read professional association newsletters less often, and those with fewer connections to colleagues may have been underrepresented.

Another shortcoming of the study was the inability to aggregate multiple questionnaire items into constructs. In other words, in most cases we could not measure the same dimension (e.g., self-efficacy) with a total score calculated over several questions. A total score would have required at least an internal consistency calculation and a normality check of the distribution of the total score. Unfortunately, there were too many situations where one or more items did not apply to the participating physician. The total number of respondents who provided a response for each item in a construct on a Likert-type scale remained very small. The situation in practice turned out to be much more diverse than we had planned.

The participants were asked to answer sociodemographic questions such as age, number of household members etc. Questions about medical specialty, years of practice, age and number of siblings were included as well. Participants had to respond to statements in which they had to choose from a six-point scale ranging from 1—Strongly Disagree to 6—Strongly Agree. This scale was chosen to avoid a neutral point in the Likert-type scale. 

In the larger study, the intention was to determine whether the responses were distributed according to the theoretical models of connection between self-efficacy, willingness to work, and duty to care. However, the nature of the data did not allow for linear regression. Most questions had a “not applicable” response option and were not normally distributed. The “not applicable” response choice was introduced in most items because of real-life situations that might not apply to the respondent. An example is item 48: “When asked to work with COVID-19 patients, I was willing to respond”. Because not all physicians were asked to work with COVID-19 patients, there was a “not applicable” response option. Another example is item 53: “I was willing to provide direct patient care even though I did not have access to a K95 mask, although I should have used it.” Many physicians had access to masks from the very beginning, and were never confronted with this situation where they had to choose whether to risk their health. We chose logistic binary regression, which can process large amounts of data and select relevant predictors for a given response.

All the 1285 participants responded to the trust in the government item: “I felt I could raise public health issues with the relevant institutions.” The mean score on this scale for the entire sample was 2.68, with a minimum of 1 and a maximum of 6. The median score of the scale was 2. To distinguish between the two groups, one more likely to agree with this statement and one more likely to disagree with this statement, we used the median and recoded a new binary variable. Scores greater than 2 (above the median) were coded as “more trust in relevant institutions” and scores equal to or less than 2 were coded as “less trust in relevant institutions”. The frequencies for the two modalities of the variable were 712 (55.4%) of respondents who had lower trust (tended to disagree) and 573 (44.6%) of respondents who had higher trust (tended to agree). Using the new dichotomous variable as the dependent variable for a binary logistic regression, we performed Backward-Wald calculations to determine the best predictors for the two categories of responses. Binary logistic regression was used because most of the variables that could potentially be considered as predictors did not have a normal distribution. Another reason was that many variables had a “not applicable” choice, which would have biased the attempts to use other types of statistical indicators.

The database we worked with contained 143 items. For a first selection, we used Mann–Whitney tests for variables measured by scales and chi-square tests for categorical variables. These showed us the variables for which there was a significant difference between the respondents in the “high confidence” category and those in the “low confidence” category. This approach resulted in 74 items for which the score was significantly different between the two categories. All were entered into a binomial logistic regression equation. The probability of removal was set at 0.05, as was the probability of introduction. Due to space limitations, we have included only those items that were removed in the last five steps of the logistic regression in Table 2.

## 3. Results

In this study, we used a binary logistic regression model to create a profile of physicians who felt that they could raise public health issues with relevant institutions during the COVID-19 pandemic as opposed to those who did not trust these institutions enough to raise public health issues. The binary logistic regression model shows several items. The responses to these items tend to predict agreement or disagreement with the item “I felt I could raise public health issues with the relevant institutions.” This way of analyzing the data can provide a profile of the sort of person who tends to agree or disagree with the above item. The Backward-Wald calculation eliminates insignificant predictors and keeps only those variables that can participate in the binary logistic model. Therefore, the results can provide insight into the mindsets of physicians who feel that they can raise public health issues with relevant institutions as opposed to those who do not feel this way.

We obtained five predictors that were able to discriminate among the respondents who tended to agree with the trust statement (felt they could raise public health issues with relevant institutions) and those who tended to disagree with the same statement. The Cox and Snell pseudo-R^2^ coefficient was 0.09 and the Nagelkerke R^2^ was 0.12. This indicates a modest agreement between the model and the real data. Because R^2^ cannot be used as a definitive or exact value for the percentage of variance accounted for, the best interpretation of this model is in terms of a modest correspondence between the data and the model resulting from the analysis. The R2 coefficients are indicators of how well the model fits the actual data. The closer the values are to 1.0, the better the model matches the actual data. Values of 0.09 and 0.12 are far from the reference value of 1.0, indicating a modest agreement between the model and the real data [16]. The model was more exact with respect to those with lower trust (73%) than those with higher trust (53%); see Table 3. The model indicates the accuracy of the classification. It shows the relative proportions of low and high confidence that the participants correctly identified. The model predicts those with low confidence with significantly higher accuracy. This means that low scores on the five predictor variables are more relevant for the low-confidence respondents than high scores on the five predictor variables are for the high-confidence respondents. From a practical point of view, this may tell us that the tendency to disagree on the items presented as predictors is more relevant for those who do not feel that they can raise public health issues with the relevant institutions. Let us consider a hypothetical case: a doctor disagrees with the statement “I trusted that I would be safe at work during the pandemic.” He also disagrees with the statements “The financial bonus we were promised justified the risk I took” and “I was trained in the use of the protective equipment used during the pandemic.” Finally, the same physician disagrees with the statements “My colleagues and I share the same values” and “Compared to before the pandemic, I enjoy my work just as much.” Based on these responses, the model predicts that in 73% of the cases such a doctor would have little confidence in raising public health issues with the relevant institutions. If another hypothetical physician had responded in agreement with the above statements, the model would have predicted that such a physician would have high confidence in raising public health issues with the relevant institutions only in 53% of cases. In other words, the predictive model is better at identifying those who are distrustful. This suggests potential strategies for identifying factors that might alter a physician’s propensity to react negatively to such items. According to the predictive model, a significant change in these items would lead to a significant change in confidence. It is important that the model is better able to predict those with low trust, as changing their perceptions is the goal of any intervention aimed at increasing trust in public institutions.

In the survey, the respondents who agreed with the statement “I felt I could raise public health issues with the relevant institutions” were more likely to agree with the following statements: (1) “I trusted that I would be safe at work during the pandemic”, (2) “The financial reward we were promised justified the risk I took”, (3) “I have been trained on the use of protective equipment used during the pandemic”, (4) “My colleagues and I share the same values”, and (5) “I enjoy my work just as much as I did before the pandemic”. Table 4 depicts each item along with the beta coefficient, standard deviation, and Wald coefficient in the final model of the logistic binary prediction. Table 2 presents the items removed in the last five steps of the backward Wald regression

Considering that all of the odds ratios have values above 1, it can be concluded that agreement on all the predictors listed in Table 4 point in the direction of agreement with the criterion variable question. In other words, agreement with each significant predictor increases the odds that the respective respondent agrees that he or she can raise public health concerns with the relevant institutions. For example, for the item “I trusted that I was safe at work during the pandemic,” interpretation of the odds ratio of 1.15 shows that the strength of the association is low. The odds of being among those who are highly confident that they can raise public health issues are 1.15 times higher for those who trusted that they felt safe at work than for those who did not trust that they felt safe at work.

## 4. Discussion

In this study, we used a binary logistic regression model to profile physicians who felt that they could raise public health issues with relevant institutions versus those who felt that they could not.

The first predictor we identified was agreement with the following statement: “I trusted that I would be safe at work during the pandemic.” This finding is consistent with the findings of Shahrabani et al. (2021) that higher levels of compliance were associated with higher levels of trust in the Ministry of Health and the healthcare system. The cited study found a positive significant relationship between trust in public institutions and personal compliance with preventive measures [7]. However, we argue that feeling safe at work can be interpreted as a result of both personal compliance with rules and effective institutional policies in providing safety to hospital and clinic workers. This finding is consistent with what Imai and colleagues (2010) found in relation to H1N1 influenza epidemics. In their study, the most important factor that motivated people to work was feeling protected by their country, local government, and hospital [8]. Therefore, it is not surprising that the physicians in our study perceived a relationship between their sense of safety at work and their confidence in raising public health issues with relevant institutions. It is noteworthy that our prediction model is more accurate for physicians who disagreed with the above statement. That is, disagreement with the statement “I trusted that I would be safe at work during the pandemic” predicted, among other variables, disagreement with the statement “I felt that I could raise public health issues with relevant institutions.” This is consistent with the findings of Zohar and colleagues (2022) that physicians with low involvement in the decision-making process during the pandemic reported lower levels of trust in politics than those with high involvement [4]. Other external factors may play a role in the association we found. First, we hypothesize that the allocation of resources during a pandemic plays an important role in ensuring the safety of health care workers. Therefore, the relationship between trust in public institutions and feeling safe at work could be explained by this variable. We might assume that effective resource allocation leads to increased safety of health care workers during an epidemic. This increased safety could in turn be associated with increased trust in the government institutions that provided this resource allocation. Second, the communication strategies of public institutions might directly influence trust in institutions. However, this influence could be achieved through a greater sense of security as well. Especially in a pandemic context such as COVID-19, communicating clear measures about the new virus could make health workers feel safer. Therefore, a greater sense of security might increase health workers’ confidence that institutions are open to their feedback on relevant public health issues. Finally, the perceived transparency of institutions might affect trust both directly and through a promoting a greater sense of security. We hypothesize that if healthcare professionals perceive a lack of transparency, this could lead them to think of dangers that are not readily disclosed by the authorities in order to avoid panic. Thus, a lack of transparency might lead to greater distrust of institutions by healthcare workers through job insecurity. It appears that trust is closely related to how well physicians feel protected by these institutions and, in turn, to a physician’s willingness to get involved and provide feedback on policy and public health issues. Low levels of trust and involvement appear to correlate with feeling less safe during an outbreak. Therefore, promoting workplace safety and finding out what makes physicians feel safe in their workplace has a good chance of increasing trust in relevant institutions.

Another significant predictor in our binary logistic model was agreement with the statement “The financial reward we were promised justifies the risk I took.” First, it is important to look at the half of physicians who disagree with the statement in the dependent variable. As we noted above, our predictive model is significantly more accurate for those physicians who tend to disagree with the statement “I felt I could raise public health issues with the relevant institutions” (73% agreement between the model and the actual data) than for those who agree with the statement (53% agreement). This means that disagreement with the statement “The financial reward we were promised justified the risk I took” predicted a tendency to disagree with the statement “I felt I could raise public health issues with the relevant institutions.” One possible interpretation of this finding is that physicians do not perceive a respectful relationship between themselves and the public institutions involved in policy making and implementation. A financial bonus may have the paradoxical effect of increasing distrust of government, as physicians may feel that duty, rather than financial incentives, drove them to participate in efforts during the pandemic. One study has shown that those who were less involved in the decision-making process had less trust in public institutions [4]. Another study found that higher levels of trust in public institutions were associated with higher levels of compliance with public health guidelines [7]. We suggest that for at least some physicians it was their sense of partnership that promoted trust and policy implementation, rather than the asymmetric relationship between employees and an employer using financial incentives. However, for those for whom the financial bonus was important, this incentive may have been interpreted as a sign of respect. It seems plausible that physicians who are distrustful of public institutions may feel that the financial reward they receive is too small compared to the effort they put in during the pandemic. The act of receiving a (possibly inadequate) financial reward for a very large extra effort has two sides. Certain physicians may perceive that governmental institutions recognize their extra effort, while other physicians may perceive that they have received a sum of money instead of real recognition for the sacrifice they have made. In certain cases, therefore, the lack of financial compensation may prove more useful than financial compensation that is disproportionately small in relation to the effort invested. When there is no form of financial reward, it leaves room for society to recognize a possible form of heroism. If there is a form of financial reward, it is more difficult for those on the sidelines to interpret the act as heroism. This could be a hypothetical mechanism for the paradoxical effect we observed. The data in our final logistic prediction model are insufficient to speculate further on the reasons for this relationship. We suggest that a financial bonus may help to promote trust, but only if it is large enough to be perceived as a respectful way of acknowledging physician’s efforts during an outbreak.

Physicians who were more likely to agree with the statement “I have been trained in the use of protective equipment to be used during the pandemic” were more likely to feel that they could address public health issues. These data are consistent with previous findings about trust between physicians and public institutions during a pandemic. Previous studies we reviewed did not look for a specific relationship between trust in institutions and receipt of training. However, there is consistent evidence of a strong relationship between trust and willingness to use government-provided tools and follow safety measures. Zohar and colleagues found that health professionals who did not trust public institutions were less likely to use government tools to track infected patients during the COVID-19 pandemic [4]. Higher levels of trust in public institutions were associated with higher levels of compliance with Ministry of Health guidelines [7]. In a study of the H1N1 outbreak, the most influential factor motivating people to work was feeling protected by their country, local governments, and hospitals [9]. It is clear that training in safety measures can be perceived by health care workers as a measure of protection and care by both hospitals and government agencies. As such training promotes trust, this can be seen in physicians’ feeling that they can raise public health concerns. We do not claim that the feeling of trust necessarily translates into the behavior of raising public health issues with relevant institutions. However, our model suggests that more trust in the government and related agencies promotes the feeling that physician can do so if needed. Of course, the reverse is important as well; there is a tendency for the respondents who disagree with having received training to feel that they cannot raise public health issues with relevant institutions. This could be explained by a feeling of “not being heard and not being cared for” that physicians might have during pandemic efforts. In conclusion, both our results and previous findings strongly suggest that training in the use of protective equipment promotes trust in both the hospital and public institutions involved in managing an outbreak. 

An interesting finding in our model is that agreement with the statement “My colleagues and I share the same values” tends to predict agreement with the outcome variable “I felt I could raise public health issues with the relevant institutions.” The sense of belonging to a professional community tends to extend a sense of trust in relevant institutions. It could be hypothesized that these public institutions are composed of the same doctors who make up the collegium of professionals. A doctor may have a greater or lesser sense of belonging to a community of professionals, and in this sense he or she may have more or less trust in public institutions that are mainly populated by the same types of professionals. This is by far one of the most interesting results. This predictor turned out to be the best in the whole model (odds ratio = 1.19). We did not find such a relationship in the previous literature; thus, further investigation may prove useful and even crucial for future situations in which societies deal with outbreaks. The reverse is true as well, in that feeling that one does not share the same values as their colleagues translates into less confidence in raising public health issues. There seems to be an important link between the trust a physician feels in the professional community as a whole and the trust he or she feels towards relevant institutions. It is more difficult to translate this finding into a specific action; however, this finding suggests that a sense of belonging to a professional community that promotes the same values may play a key role in a physician’s trust in relevant institutions. As a psychotherapist, one of the authors of this article suggests that specific medical events and conferences should be organized to bring together members of the Directorate of Public Health, physicians working in hospitals and private practices, and members of the Ministry of Health. These events should be held on a regular basis to ensure a sense of belonging and shared values over time The corporate environment has brought the idea of team-building to Romania. It is well known that the purpose of such events is to strengthen the feeling of belonging to a team, with the aim of improving the quality of communication and the ability to work as a team. On different sides of imaginary barriers, doctors can forget that they share the same values, and in a crisis situation, such as the COVID-19 pandemic this can bring to light possible negative images that doctors may have of public institutions. Case studies can play a special role in various types of training. Learning and team-building situations can be designed in which teams of five people might work on a case study of the response to an epidemic. Team members would be recruited from doctors in hospitals or clinics and from specialists in government institutions. The case could be thought of in such a way that the whole team is in the position of a crisis cell which has to coordinate communication with the public, communication with hospitals, implementation of public health measures, budget optimization, etc. Thus, the team members are confronted with the different types of challenges that arise in such situations. They can realize that the whole team is trying to maximize the good for society as a whole. Each team member would have the opportunity to realize that he or she has similar or identical values to their colleagues, and that the urgency of a situation is not a prerequisite for the abandonment of these values. On the contrary, teams of people seek the best way to achieve an optimal outcome. Ultimately, this should translate into greater trust during the hardships of disease outbreaks, as individuals and groups will have cemented a degree of mutual respect and a sense of shared values.

Finally, agreement with the statement “Compared to before the pandemic, I enjoy my work just as much” tended to predict agreement with the outcome variable “I felt I could raise public health issues with the relevant institutions.” This finding is consistent with the positive and highly significant correlation found by Shahrabani and colleagues between trust and positive emotions (r(198) = 0.24; *p* < 0.01) and the even stronger and negative correlation between trust and negative emotions (r(198) = −0.34; *p* < 0.01) [7]. This can be interpreted as a link between good working conditions and trust. However, general positive feelings in the workplace seem to influence trust as a positive emotion in itself. Feelings of helplessness and hopelessness often go hand in hand, as do feelings of control and job satisfaction. It is difficult to say how much of the relationship between trust and positive emotions can be explained by personality traits, the nature of physician–institution interactions, and other contextual factors. However, it is likely that fostering general positive feelings and better working conditions plays a role in the trust physicians feel in institutions. Working conditions do matter and policymakers should strive to improve them, especially when the system is dealing with an outbreak that is taxing everyone’s efforts.

Despite the large number of respondents, there are several limitations to this study. An obvious limitation of the study is that the Cox and Snell pseudo R^2^ and Nagelkerke R^2^ coefficient values were 0.09 and 0.12, respectively. This means that our results can mainly be used as indicators of trends rather than clear differences between categories of people. These R^2^ values mean that the model explains at most 12% of the variance of the criterion variable. This proportion is low and may indicate that other variables can be found to explain more of the variance of the criterion variable. The modest fit of the model to the actual data suggests that cautious interpretation of the results is appropriate. The study captured a relatively weak relationship between the predictors and the criterion. This relationship may serve more as a suggestion for further research. Furthermore, the validity of the approach may be affected by possible confounding variables. Because a correlational relationship does not imply a causal relationship, other variables may be found to be related to both the predictor variables and the criterion variable. Such relationships would require, for example, structural equations that capture moderation and mediation phenomena. Therefore, the conclusions of this paper should be generalized with caution. The findings in this paper need to be supported by other research that uncovers relationships between trust and the variables we have mentioned. In addition to this limitation, it should be noted that the criterion variable in this study was measured by a single item. The responses to this question were divided into two categories by the median test, resulting in a dichotomous variable. Further research may identify other relevant predictors that explain more of the variance in the criterion variable. At the same time, a more precise measure of the level of confidence physicians have in public institutions to address public health issues during epidemics could be obtained. This more precise measurement could be achieved by using multiple items that refer to the same construct, e.g., physicians’ trust in public institutions.

Another limitation of the survey is the “not applicable” option in the answer to most questions. As explained in the Methods section, this option did not illustrate a neutral response; rather, it was introduced because, from a practical point of view, certain situations might not apply to the respondent. For example, doctors might have answered “not applicable” to the question “My family was prepared to manage without me while I worked during the pandemic” because they were away from their families anyway. For the question “My employer and I share the same values,” doctors might have answered “not applicable” because they were self-employed. For this reason the statistical processing program included only 902 cases in the analysis, i.e., 70.2% of all the respondents. The number of missing cases was 383. Therefore, the results apply to the 902 cases that were included in the analysis and not to the total of 1285. From this point of view, the statistical power and representativeness of the sample may have been diminished.

As noted in the Methods section, selection bias in the self-selection of the participants remains the most significant limitation of this study, despite the use of anonymous surveys to counteract its effects. Busy or dissatisfied physicians may have been underrepresented in the responses. In addition, the distribution of surveys through professional associations and networks may have introduced a bias towards physicians with better connections within these networks. Internet use, frequency of newsletter reading, and professional connections may have influenced participation rates. Another limitation was the inability to aggregate questionnaire items into constructs such as self-efficacy. Due to numerous instances in which survey questions might not have applied to the respondent, the total number of respondents per construct was small.

The study was conducted on a Romanian sample. There are two difficulties regarding the possible generalization of the results. First, there is the rather low power of the model to make predictions even within the context of the Romanian physician population. The R-squared coefficients are low, and there are possible confounding variables, as already mentioned in the Discussion section. Moreover, while the representativeness of the sample for the whole population of Romanian doctors is good, it is not perfect. Second, because the study involved only a group of Romanian doctors, we cannot claim generalizations to doctors in other countries. Different cultures may perceive the factors influencing medical staffs’ trust in government institutions quite differently. Only by finding the same relationships in other cultures can we hypothesize about Western doctors’ trust in government institutions. Other studies might use standardized questionnaires. These questionnaires can be adapted in other cultures to test the same set of hypotheses. Only in this way can we build general theoretical models for Western physicians’ trust in government institutions in a pandemic situation.

Aside from these limitations, the study worked with a large number of observations and used a safer method of computing data against false positives, which are often found in parametric statistical analysis.

## 5. Conclusions

In this study, we used a binary logistic regression model to profile physicians who felt that they could raise public health issues with relevant institutions versus those who felt that they could not. It is important to note that a predictive model can only identify possible relationships between variables. Obviously, no correlation system can claim to identify cause-and-effect relationships. As with any study of this nature, the results must be interpreted with caution. Our findings can only suggest new directions for research and possible interventions. These suggestions may be confirmed by other findings with more controlled research designs. Our results suggest that physicians who trusted the system sufficiently to raise public health issues with relevant institutions were more likely to feel that they shared the same values as their colleagues, report that they were trained in the use of protective equipment used during the pandemic, trust that they were safe at work during the pandemic, enjoy their work as much as before the pandemic, and feel that the financial bonus they received justified the risk. In contrast, the results suggest that physicians who felt unable to raise public health issues with relevant institutions during the pandemic were more likely to feel that they did not share the same values as their colleagues, to disagree that they had received training in the use of protective equipment, to disagree that they felt safe at work during the pandemic, and to disagree that they enjoyed their work as much as before the pandemic. Our results are consistent with other findings about physicians’ trust in institutions during outbreaks. Despite low claims of generalizability, our study suggests that certain actions are needed. To cultivate trust in institutions, even before an epidemic breaks out, policymakers could focus on making health care workers feel safe at work, fostering a sense of belonging to the profession, cultivating the intrinsic rewards of the medical profession, training doctors in the use of protective equipment, and providing significant bonuses for higher-risk situations.

As shown above, the use of logistic regression in this study had both advantages and disadvantages. The logistic regression equation allowed for the processing of a large number of variables and avoided false positives through the nonparametric approach. However, the associations identified were modest and could not explain any of the possible causal relationships. An alternative approach would be to construct questionnaires with general statements about trust in public institutions (e.g., “I believe that public institutions will tell the truth when implementing health care policies during a pandemic”). These general statements would no longer require a “not applicable” option. Several such items could be aggregated into constructs with theoretical validity and internal consistency. A normal distribution of the scores would allow either linear regressions or structural equations. The latter could indicate a possible theoretical explanatory model much better than logistic regression.

This study has other limitations that must be considered. The most important limitation is the potential for self-selection bias on the part of the participants. Another limitation is that the surveys were distributed through professional associations and networks, resulting in a possible bias toward better-connected physicians. The study design did not permit us to aggregate multiple questionnaire items into constructs, resulting in single-item measures. The pseudo-R2 coefficient values are low, indicating that the model can explain at best 12% of the variance in the criterion variable. The study may be affected by confounding the variables. In addition, the criterion variable was measured by a single item and the study had a high number of missing cases. As the study was conducted in Romania, the generalizability of the results to other cultures or countries may be limited. Therefore, this paper emphasizes the need for further research in order to establish relationships between trust and the variables mentioned. This can be done by using standardized questionnaires that can be adapted to different cultures; in this way, general theoretical models of physicians’ trust in government institutions in a pandemic situation could be developed.

Not all of the factors we identified can be directly addressed by government policy. If the results of our study prove to be valid, direct steps could be taken to increase physician confidence. Our findings suggest that as long as physicians feel that the financial reward justifies the risk, physician confidence will increase. A challenge in this regard would be to identify the point at which a financial reward “justifies” the perceived risk to the physician, given budgetary constraints. Second, training in the use of safeguards can be promoted by the central authorities. This training can be provided on a regular basis as part of a program to ensure responsiveness to pandemics. Government institutions can ensure that such programs are conducted either in hospitals or in professional schools. However, government institutions cannot guarantee the quality of the training programs unless the trainers come from the center. The latter is more difficult to implement in practice. The subjective feeling of being safe at work can depend on many factors, and not all of these factors can be controlled by government institutions or the institution where a doctor works. The most difficult to influence are the sense of shared values between the physician and colleagues and the intrinsic emotional rewards of practicing medicine. While programs can be created to cultivate a sense of belonging to a shared value system, finding ways to increase intrinsic emotional rewards seems difficult. Many variables related to the physician’s personality and value system come into play. Obviously, part of a physician’s confidence that he or she can raise public health issues with the appropriate institutions will be influenced by deeply personal variables.

The results of this study can be discussed from an ethical perspective. Institutions have a responsibility to create an environment that encourages physician feedback. In return, medical staff implement public health policies and measures. Only through this “contract” is the patient truly protected. One of the physician’s duties is to advocate for his or her patients. Having the confidence to raise public health issues with the relevant institutions is essential in order for a physician to perform his or her job. If further studies confirm the importance of the factors we have identified in this paper, patient care may be indirectly improved by influencing these factors.

## Figures and Tables

**Table 1 healthcare-11-01736-t001:** Distribution of questionnaire respondents by historical areas of the country and types of communities.

	Place of Medical Practice				Total
	Village	Small Town (10,000–100,000 Inhabitants)	Medium-Sized City (100,000–500,000 Inhabitants)	Big City (over 500,000 Inhabitants)	
Transilvania	22	90	112	74	298
Banat	4	8	25	32	69
Crișana	6	11	29	8	54
Moldova	57	41	74	136	308
Oltenia	11	30	39	19	99
Muntenia	28	37	44	256	365
Bucovina	8	22	9	1	40
Maramureș	0	5	12	1	18
Dobrogea	2	9	8	15	34
Total	138	253	352	542	1285

**Table 2 healthcare-11-01736-t002:** Items removed in the last five steps of the backward Wald regression.

Item in the General Questionnaire	Score	Sig.
27. Years of medical practice	0.827	0.363
45. A young healthcare professional should work on the front line during a pandemic.	0.681	0.409
59. My role in the overall response to the pandemic was an important one.	0.569	0.451
70. Not showing up for work during the pandemic means abandoning one’s team.	2.467	0.116
71. Not showing up for work during a pandemic means abandoning one’s patients.	0.319	0.572

**Table 3 healthcare-11-01736-t003:** Predictive model of the binary logistic regression.

Predictive Model			
	Lower Trust (Predicted)	Higher Trust (Predicted)	Percentage Correct
lower trust (expected)	352	131	72.9
higher trust (expected)	198	221	52.7
Overall percentage			63.5

**Table 4 healthcare-11-01736-t004:** Items with significant prediction power in the binary logistic regression model.

Item in the General Questionnaire	B Coefficient	Standard Deviation	Wald	Odds Ratio	Sig.
I trusted that I was safe at work during the pandemic.	0.14	0.05	8.78	1.15	<0.01
The financial reward we were promised justifies the risk I took.	0.08	0.04	4.59	1.09	0.03
I have been trained on the use of protective equipment used during the pandemic.	0.16	0.04	19.46	1.17	<0.01
My colleagues and I share the same values.	0.17	0.06	9.16	1.19	<0.01
I enjoy my work just as much as I did before the pandemic.	0.11	0.05	6.09	1.12	0.01
Constant	−2.55	0.32	64.10	0.08	<0.01

## Data Availability

The data presented in this study are available on request from the corresponding author.

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
