# Peer review of "Physicians’ Trust in Relevant Institutions during the COVID-19 Pandemic: A Binary Logistic Model"

_healthcare, 2023, doi:10.3390/healthcare11121736_

Round 1
Reviewer 1 Report
The paper appears to be well-written. The study is relevant as it investigates an important issue - the perception of health professionals towards institutions and governments during epidemics. The authors aimed to create a profile of physicians who feel comfortable raising public health issues during a pandemic, which can provide valuable insights into how to improve public health response. In terms of originality, while the topic has been previously addressed in the literature, the study is still original in its approach in Romania, where research in this area has been limited. The authors provided appropriate justification about choosing binary logistic regression as a statistical method to identify predictors. The findings have practical implications for public health officials in Romania and other countries facing similar challenges.
I’d recommend the authors to consider the following:
1. Abstract: to remove the structured sub-heading “(1) Background:” and so on.
2. Although the authors described the purpose (line 126), it seems not very clear in what way the current study could contribute to the literature gap. I’d recommend the authors to add more specific hypothesis, research questions or conceptual frameworks here.
3. The authors mentioned to avoid using a neutral point in the Likert scale (line 147), however it allowed “not applicable” (line 150) which seems to cause neutrality and missing values to the data set. I’d appreciate if the authors could explain the intention of such a designing here and how to treat the “not applicable” value. I suppose these questions look for opinion rather than confirming a fact, the “not applicable” might be close in meaning with a “neutral” rather than a missing value.
4. Method: It could be better to describe the structure of the tools used: Part 1: Demographic, Part 2: Statements about …, to specify number of items or scales used etc.
5. Please specify the statistical software used to make the statistics.
6. Results: Line 184: “The Cox and Snell pseudo R2” seems to be a single coefficient, but the authors provided two values (0,09 and 0,12) – Line 185. I supposed these values to be 0.09 and 0.12 with decimal. I guess the 0.12 could be the McFadden’s R2 or Nagelkerke R2, please check. I highly recommend the author to specify the references used for calculating or interpreting these values (Eg. https://thestatsgeek.com/2014/02/08/r-squared-in-logistic-regression/, https://web.pdx.edu/~newsomj/cdaclass/ho_logistic.pdf etc. )
7. The Cox & Snell R Square = 0.09 implies that the model explains 9% of the variance in the dependent variable, I’d recommend noting (in the Discussion or Limitation section) that these values appear to be relatively low, indicating that there may be other variables not included in the model that could explain more of the variance in the dependent variable. Therefore, additional research may be needed to identify other relevant predictors of the dependent variable.
8. Table 1 (Line 188). The sum of samples participated in the model in this table is 902, quite lower than the total sample size of 1285. I’d recommend the authors to mention/discuss this fact in the Results or Discussion section.
9. Table 2 (Line 197). It could be more convincing if the authors could show the items that were eliminated (the variables not in the equation), so that there will be a picture about the other factors that found insignificant in this data set, but still might contribute to the dependent variable.
Author Response
- Recommendations from the first reviewer
- We removed the structured sub-heading of the Abstract
- Although the authors described the purpose (line 126), it seems not very clear in what way the current study could contribute to the literature gap. I’d recommend the authors to add more specific hypothesis, research questions or conceptual frameworks here.
We developed a paragraph starting on line 175 in the manuscript:
"The literature shows that both public trust and health care professionals' trust in public institutions increase the likelihood of effective health care policy implementation. This is even more true during epidemics. Studies to date have measured various factors that may contribute to this trust: the use of experts, the promotion of credible sources, the public's belief that professionals are genuinely involved in the decision-making process, and the professionals' own belief that they are involved in the decision-making process. These have increased trust in evidence-based science. This promotes a sense that the health worker is protected. It also encourages the avoidance of biases promoted by the media. It fosters trust in the integrity and ability of institutions to implement effective policies. We are still far from developing a coherent theoretical model that takes all these factors into account. This paper aims to expand the list of factors that may influence health care professionals' trust in public institutions. Professionals’ trust in government agencies has been less studied than public trust. This paper contributes to the literature gap on factors influencing health care professionals' trust in public institutions implementing health policies. The more we know about the factors that influence health care professionals' trust, the more we can influence these factors in a future epidemic. Because the context of an epidemic requires rapid intervention, knowing the factors in advance can pave the way for a rapid response. Changing the factors that influence physicians' trust in government institutions can be done early. If all these trust interventions prove effective, physicians will be more likely to respond favorably to public health interventions when the need arises".
- The authors mentioned to avoid using a neutral point in the Likert scale (line 147), however it allowed “not applicable” (line 150) which seems to cause neutrality and missing values to the data set. I’d appreciate if the authors could explain the intention of such a designing here and how to treat the “not applicable” value. I suppose these questions look for opinion rather than confirming a fact, the “not applicable” might be close in meaning with a “neutral” rather than a missing value.
We changed the paragraph starting on line 321 of the manuscript:
"In the larger study, the intention was to see if the responses were distributed according to the theoretical models of connection between self-efficacy, willingness to work and duty to care. However, the nature of the data did not allow for linear regression. Most questions had a "not applicable" response option and were not normally distributed. The "not applicable" response choice was introduced in most items, because there were real life situations that did not apply to the respondent. One example is item 48 " When asked to work with COVID-19 patients, I was willing to respond". Since not all physicians were asked to work with COVID-19 patients, there was a "not applicable" response option. Another example is item 53 "I was willing to provide direct patient care even though I did not have access to a K95 mask, although I should have used it.”. Many physicians had access to masks from the very beginning, so they were not confronted with this situation where they had to choose whether to risk their health. We chose logistic binary regression which can process large amounts of data and to select relevant predictors for a given response".
- Method: It could be better to describe the structure of the tools used: Part 1: Demographic, Part 2: Statements about …, to specify number of items or scales used etc.
We have added a paragraph starting on line 225 of the manuscript:
"The questionnaire, administered online, was structured in the following parts. From question 1 to question 18 both socio-demographic items (e.g. gender, age, region in Romania where the doctor works, marital status) and items related to socio-economic status (number of people with whom the doctor lives, number of elderly people in the household, number of young children, number of rooms in the household, self-assessment of economic status on a scale from 1 to 10, whether the doctor is currently still employed, etc.) were included. Question 19 targeted self-reporting of the various diseases the doctor suffers from. Because, in the online questionnaire, the respondent had to tick from a list of multiple options, in the database we unpacked each option into a separate dichotomous variable (e.g. question 19t "Does the medical professional self-report bronchial asthma, including allergic asthma" - answer YES = 1 or NO = 0). Without this decomposition into dichotomous variables, it would have been impossible to process the information resulting from item 19. Further, from item 20 to item 25, questions were asked about the medical professional's specialty and self-assessment of health status in the last two weeks, respectively in the two weeks before the pandemic. Question 26 targeted the specific place of work of the health care professional (e.g. maternity, intensive care, day hospital, etc.). Because two or more of these options could be present at the same time, we split each option into a separate dichotomous variable, similar to item 19. From question 27 to question 36, the healthcare professional was asked whether he/she worked in a quarantined locality, the average number of hours worked per week during the pandemic, whether he/she was hospitalized as a result of a COVID infection, etc. Question 37 referred to the types of activities in which the healthcare professional was involved during the pandemic (e.g. collecting biological samples from suspected COVID-19 patients, performing aerosol-generating procedures, examining a patient with respiratory symptoms, etc.). This question was also broken down into several dichotomous variables, since the same physician performed one or more of these activities. Question 99 asked for a ranking from 1 (most important) to 3 factors that negatively influenced medical practice (e.g. infection control procedures, discharge of non-emergency patients from hospital, etc.). Questions 129 and 130 also asked for a ranking of main sources of information (e.g. direct supervisors, other colleagues in the community, etc.) and main supportive factors when the doctor faced a professional problem (e.g. colleagues at work, professional organizations, etc.). Questions 99, 129 and 130 were in turn broken down into separate variables for each option, where the number corresponding to the hierarchy (1, 2 or 3) was recorded. The remaining questions in the range 38 - 144 were questions answered on 6-point Likert-type scales and referred to various constructs in the literature: self-efficacy, duty, willingness to work, institutional reciprocity and government reciprocity. Due to the lengthy process of constructing the questionnaire, the items were not grouped in the database according to the construct they were part of. However, each item was labelled as such in the database (e.g. Question 49: "I was aware of my role during the pandemic" - Self-efficacy; Question 43: "A healthcare professional who does not have children should work on the front lines during a pandemic" (reversed item) - Duty). After the data collection, due to multiple situations that did not apply to one or more of the physicians, the research team abandoned the aggregation of items into the questionnaires by calculating the internal consistency coefficient and a total per scale. All the data were processed in IBM SPSS Statistics 20.0".
- Please specify the statistical software used to make the statistics.
We specified the software in one sentence, line 269: "IBM SPSS Statistics 20.0".
- Results: Line 184: “The Cox and Snell pseudo R2” seems to be a single coefficient, but the authors provided two values (0,09 and 0,12) – Line 185. I supposed these values to be 0.09 and 0.12 with decimal. I guess the 0.12 could be the McFadden’s R2 or Nagelkerke R2, please check. I highly recommend the author to specify the references used for calculating or interpreting these values (Eg. https://thestatsgeek.com/2014/02/08/r-squared-in-logistic-regression/, https://web.pdx.edu/~newsomj/cdaclass/ho_logistic.pdf etc. )
We have corrected and added an explanation starting on line 380 of the manuscript:
"The Cox and Snell pseudo R2 coefficient was 0.09 and the Nagelkerke R2 was 0.12. This indicates a modest agreement between the model and the real data. Since R2 cannot be used as a definitive or exact value for the percentage of variance accounted for, the best interpretation of this model is in terms of a modest correspondence between the data and the model resulting from the analysis. The R2 coefficients are indicators of how well the model fits the actual data. The closer the values are to 1.0, the better the model matches the actual data. Values of 0.09 and 0.12 are far from the reference value of 1.0, indicating a modest agreement between the model and the real data16".
We added a 16th reference, which is the source we used to create and interpret the logistic regression:
Howitt, D, Cramer, D. Understanding Statistics in Psychology with SPSS, Seventh Edition, Pearson Education Limited, Harlow,Harlow, UK, 2017. pp.647-662
- 7. The Cox & Snell R Square = 0.09 implies that the model explains 9% of the variance in the dependent variable, I’d recommend noting (in the Discussion or Limitation section) that these values appear to be relatively low, indicating that there may be other variables not included in the model that could explain more of the variance in the dependent variable. Therefore, additional research may be needed to identify other relevant predictors of the dependent variable.
We have improved and expanded the paragraph starting on line 608 of the manuscript:
"Despite the large number of respondents, there are several limitations to this study. An obvious limitation of the study is that the Cox and Snell pseudo R2 and Nagelkerke R2 coefficient values were 0.09 and 0.12, respectively. This means that our results can mainly be used as indicators of trends rather than clear differences between categories of people. These R2 values mean that the model explains at most 12% of the variance of the criterion variable. This proportion is low and may indicate that other variables can be found to explain more of the variance of the criterion variable. The modest fit of the model to the actual data suggests a cautious interpretation of the results. The study captured a relatively weak relationship between the predictors and the criterion. This relationship may serve more as a suggestion for further research. Furthermore, the validity of the approach may be affected by possible confounding variables. Because a correlational relationship does not imply a causal relationship, other variables may be found to be related to both the predictor variables and the criterion variable. Such relationships would require, for example, structural equations that capture moderation and mediation phenomena. Therefore, the conclusions of this paper should be generalized with caution. The findings in this paper need to be supported by other research that uncovers relationships between trust and the variables we have mentioned. In addition to this limitation, it should be noted that the criterion variable in this study was measured by a single item. The responses to this question were divided into two categories by the median test, resulting in a dichotomous variable. Further research may identify other relevant predictors that explain more of the variance in the criterion variable. At the same time, a more precise measure of the level of confidence physicians have in public institutions to address public health issues during epidemics could be obtained. This more precise measurement can be achieved by using multiple items that refer to the same construct (e.g., physicians' trust in public institutions) ".
- Table 1 (Line 188). The sum of samples participated in the model in this table is 902, quite lower than the total sample size of 1285. I’d recommend the authors to mention/discuss this fact in the Results or Discussion section.
We added the following paragraph starting on line 637 of the manuscript:
"Another limitation of the survey is the "not applicable" option in the answer to most questions. As explained in the method section, this option did not illustrate a neutral response. It was introduced because, from a practical point of view, some situations did not apply to the respondent. For example, to the question "My family was prepared to manage without me while I worked during the pandemic", some doctors answered "not applicable", as they were away from their families anyway. To the question "My employer and I share the same values", some doctors answered "not applicable" because they were self-employed. From this, it resulted that the statistical processing program included only 902 cases in the analysis, i.e. 70.2% of all the respondents. The number of missing cases was 383. Therefore, the results apply to the total of 902 cases that were included in the analysis and not to the total of 1,285. From this point of view, the statistical power and representativeness of the sample may have been diminished".
- Table 2 (Line 197). It could be more convincing if the authors could show the items that were eliminated (the variables not in the equation), so that there will be a picture about the other factors that found insignificant in this data set, but still might contribute to the dependent variable.
Added paragraph starting on line 354 to the manuscript:
"The database we worked with contained 143 items. For a first selection, we used Mann-Whitney tests for variables measured by scales and chi-square tests for categorical variables. These showed us the variables for which there was a significant difference between the respondents in the “high confidence” category and those in the “low confidence” category. This resulted in 74 items for which the score was significantly different between the two categories. All were entered into a binomial logistic regression equation. The probability of removal was set at 0.05 and the probability of introduction was also set at 0.05. Due to space limitations, we included the items that were removed in the last five steps of the logistic regression in Table 4".
Included Table 4: Items that were removed in the last five steps of the Bacward-Wald regression.
|
Item in the general Questionnaire |
Score |
Sig. |
|
27. Years of medical practice |
.827 |
.363 |
|
45.A young healthcare professional should work on the front line during a pandemic. |
.681 |
.409 |
|
59. My role in the overall response to the pandemic was an important one. |
.569 |
.451 |
|
70.Not showing up for work during the pandemic means abandoning one's team. |
2.467 |
.116 |
|
71.Not showing up for work during a pandemic means abandoning one's patients. |
.319 |
.572 |
Reviewer 2 Report
Introduction
The introduction lacks a clear and concise research objective or research question. It would be beneficial to explicitly state the purpose of the study and what the researchers aim to investigate or contribute to the existing literature.
While the introduction briefly mentions the significance of trust in public health policies, it would be helpful to provide more specific examples or evidence to support this claim. Additionally, providing statistics or data on the impact of public trust on compliance with health measures could strengthen the argument.
The introduction mentions previous studies conducted in Israel and Japan, but it would be useful to highlight how these studies are related to the current research and how they have informed the research objectives or hypotheses of the present study. Connecting the previous studies to the gaps in the literature or research questions of the current study would provide more context and justification for the research.
The introduction briefly mentions the potential vicious cycle between mistrust, non-compliance, hardship, and further mistrust during public health crises, but it would be beneficial to elaborate on this point and explain how it relates to the current study. Clarifying the specific mechanisms or factors that contribute to the cycle of mistrust and its implications for public trust in institutions would enhance the introduction's impact.
While the introduction effectively highlights the importance of professionals' trust in government agencies and the need for further research in this area, it would be helpful to outline the specific gaps or knowledge deficits that the current study aims to address. This would provide a clearer rationale for conducting the research and contribute to the significance of the study.
Literature Review:
While the introduction references several studies, it lacks a critical analysis and synthesis of existing literature on trust in public health interventions. Provide a more comprehensive review of relevant studies, identifying gaps and inconsistencies, to demonstrate the need for further research.
Consider integrating contrasting perspectives or theories to present a balanced view of the topic and enhance the scholarly discussion.
Research Gap and Objectives:
Clearly state the research gap that this study aims to address within the context of trust in public health interventions. This will help establish the rationale for conducting the research and provide a clear focus for the study.
Explicitly state the specific objectives or research questions that will guide the study. This will enhance the clarity of the research goals and align the introduction with the subsequent sections of the manuscript.
Methodological Approach:
The introduction would benefit from a brief overview of the methodology to be employed in the study. Include information about the research design, sample selection, data collection methods, and analytical techniques to give readers a sense of the empirical approach used in the research.
Language and Style:
Pay attention to sentence structure and grammar to improve the overall readability and clarity of the text. Some sentences are overly complex and require simplification for better comprehension.
Overall Implications:
While the introduction touches on the implications of the research, it would be valuable to explicitly state the potential contributions and practical significance of the study. Clearly outline how the findings of the research can inform public health interventions and policy-making processes.
Methods
The method section provides a clear overview of the study design and data collection process. However, there are a few aspects that require further clarification and improvement.
Sample Selection: The method mentions that the sample consisted of 1285 Romanian doctors, but it does not provide information on the sampling technique employed. It is essential to explain how the participants were recruited to ensure the representativeness of the sample and generalizability of the findings. Additionally, information on the response rate and any potential biases in participant selection should be addressed.
Survey Instrument: The method briefly mentions the inclusion of socio-demographic questions and statements rated on a six-point scale. However, it lacks details regarding the specific items used in the survey. It is crucial to provide a clear description of the survey questions, including their content and intended purpose. Additionally, information on the validity and reliability of the survey instrument should be included to ensure the robustness of the data collected.
Data Analysis: While the method mentions the use of logistical binary regression, there is limited explanation of the specific steps and procedures followed in the analysis. It is essential to provide a more detailed description of the regression model, including the independent variables considered, the criteria for variable selection (Backward-Wald computations), and any assumptions or limitations associated with the analysis.
Ethical Considerations: The method section does not mention anything about ethical considerations, such as informed consent and data confidentiality. It is essential to address how ethical guidelines were followed throughout the study, including obtaining informed consent from participants and ensuring the anonymity and confidentiality of their responses.
Limitations: The method section should acknowledge the limitations of the study. For example, potential sources of bias, such as self-reporting bias or non-response bias, should be discussed. Additionally, any constraints related to the study design or data collection process should be acknowledged to provide a comprehensive understanding of the study's scope and potential implications.
Results
The results section lacks crucial details and statistical information, which hinders the understanding and interpretation of the findings. The absence of specific variables and their coefficients or odds ratios in the logistic binary regression model limits the ability to assess the significance and magnitude of the predictors. Providing this information is essential for readers to evaluate the strength and direction of the relationships.
Additionally, the discussion of pseudo R2 coefficients lacks interpretation and context. Simply stating the values without explaining their meaning and implications diminishes the usefulness of these measures. Moreover, the assertion that the model was more accurate for respondents with lower trust than those with higher trust is mentioned without supporting evidence or further explanation. This discrepancy in model accuracy should be explored and discussed in more depth.
Furthermore, the presentation of the predictors and their corresponding statements is insufficient. Merely listing the statements without providing the associated results or discussing their implications limits the reader's ability to fully comprehend the findings. A tabular format displaying the predictors, coefficients, standard deviations, and Wald coefficients would enhance the clarity and accessibility of the results.
To improve the quality of the results section, it is crucial to include specific variables and their coefficients or odds ratios in the logistic binary regression model. Additionally, providing a more comprehensive interpretation of the pseudo R2 coefficients and exploring the discrepancy in model accuracy would enhance the overall analysis. Finally, presenting the predictors and their associated statements in a tabular format with relevant statistical information would improve the clarity and comprehensibility of the findings.
Discussion
The interpretation of the Cox and Snell pseudo R2 coefficients could be further discussed. While the modest fit between the model and the actual data is acknowledged, the implications of this modest fit on the generalizability and reliability of the findings should be addressed. Additionally, the limitations of using binary logistic regression in capturing complex relationships and potential confounding factors should be acknowledged.
The discussion on the relationship between feeling safe at work and trust in relevant institutions could benefit from a more nuanced analysis. While it is suggested that feeling safe at work may be a result of personal compliance and effective institutional policies, it would be valuable to explore the potential role of external factors such as resource allocation, communication strategies, and transparency in fostering trust.
The interpretation of the relationship between financial bonuses and trust in government institutions could be further elaborated. The authors mention that the financial bonus may have a paradoxical effect on trust, but it would be beneficial to delve into the underlying reasons for this effect. Additionally, the potential influence of other factors, such as the perceived fairness of the bonus distribution and the overall support provided to healthcare workers, should be considered.
The discussion on the importance of shared values among physicians and its impact on trust in relevant institutions is intriguing. However, the practical implications and feasibility of organizing specific events to promote a sense of belonging and shared values need further exploration. The authors could provide more details on how such events could be structured and the potential challenges that may arise in implementing them.
The limitations of the study are briefly mentioned but could be expanded upon. The potential sources of bias, such as self-reporting and the reliance on a specific sample of Romanian doctors, should be acknowledged. Additionally, the generalizability of the findings to other contexts and healthcare systems should be addressed.
Conclusion
The conclusion does not adequately discuss the limitations of the study. It is important to acknowledge any potential biases, limitations in data collection or analysis, and the generalizability of the findings. This would provide a more balanced interpretation of the results and acknowledge the study's shortcomings.
The conclusion does not clearly establish causality between the identified predictors and the ability of physicians to raise public health issues. While the logistic regression model identifies associations, it is crucial to emphasize that correlation does not imply causation. The conclusion should exercise caution in making definitive claims about the factors influencing physicians' trust and ability to raise concerns.
The conclusion lacks a critical reflection on the methodology used. It would be valuable to discuss potential alternative approaches or limitations of using binary logistic regression as the primary analytical method. Additionally, suggestions for future research directions or methodological improvements could be included to encourage further investigation.
The conclusion does not address potential ethical considerations. It is important to discuss the ethical implications of the findings, such as the potential impact on patient care, the duty of physicians to advocate for public health, and the responsibility of institutions to foster an environment conducive to raising concerns.
The conclusion could benefit from a more nuanced discussion of the implications for policy and practice. Merely suggesting that policymakers focus on specific factors without delving into the challenges and practicalities of implementing these recommendations may limit the usefulness of the findings.
Author Response
- Recommendations from the second reviewer
Introduction
- The introduction lacks a clear and concise research objective or research question. It would be beneficial to explicitly state the purpose of the study and what the researchers aim to investigate or contribute to the existing literature.
We added a paragraph starting on line 175 of the manuscript:
"The literature shows that both public trust and health care professionals' trust in public institutions increase the likelihood of effective health care policy implementation. This is even more true during epidemics. Studies to date have measured various factors that may contribute to this trust: the use of experts, the promotion of credible sources, the public's belief that professionals are genuinely involved in the decision-making process, and the professionals' own belief that they are involved in the decision-making process. These have increased trust in evidence-based science. This promotes a sense that the health worker is protected. It also encourages the avoidance of biases promoted by the media. It fosters trust in the integrity and ability of institutions to implement effective policies. We are still far from developing a coherent theoretical model that takes all these factors into account. This paper aims to expand the list of factors that may influence health care professionals' trust in public institutions. Professionals’ trust in government agencies has been less studied than public trust. This paper contributes to the literature gap on factors influencing health care professionals' trust in public institutions implementing health policies. The more we know about the factors that influence health care professionals' trust, the more we can influence these factors in a future epidemic".
- While the introduction briefly mentions the significance of trust in public health policies, it would be helpful to provide more specific examples or evidence to support this claim. Additionally, providing statistics or data on the impact of public trust on compliance with health measures could strengthen the argument.
We have included more detailed evidence from the cited studies:
[line 37] For instance, in a Korean study, trust in government information was identified as a primary determinant of willingness to use a mobile application for public health risk communication.
[line 41]. A study conducted in Liberia found that respondents who expressed low trust in government were much less likely to take precautions against the Ebola virus in their homes. They were also less likely to comply with government-mandated social distancing mechanisms designed to limit the spread of the virus. They were also much less likely to support potentially controversial control measures, such as "safe burial" of infected bodies.
[line 79]. For instance, Matthew Bennett argued that the success of the public health response to the COVID-19 pandemic depended on public trust in experts. According to this author's argument, when public policy claims to follow science, citizens are asked, not only to believe what experts tell them, but also to follow experts' recommendations. Bennett argues that this requires a more sophisticated form of trust, which he calls recommendation trust. The conditions for recommendation trust are more demanding than the conditions for epistemic trust. His conclusion is that many of the measures that have been proposed to cultivate trust in experts do not give the public good reasons to trust in expert-led policy.
- The introduction mentions previous studies conducted in Israel and Japan, but it would be useful to highlight how these studies are related to the current research and how they have informed the research objectives or hypotheses of the present study. Connecting the previous studies to the gaps in the literature or research questions of the current study would provide more context and justification for the research.
Paragraph starting line 186 better links the literature review to the purpose of the study: "This paper aims to expand the list of factors that may influence health care professionals' trust in public institutions".
- The introduction briefly mentions the potential vicious cycle between mistrust, non-compliance, hardship, and further mistrust during public health crises, but it would be beneficial to elaborate on this point and explain how it relates to the current study. Clarifying the specific mechanisms or factors that contribute to the cycle of mistrust and its implications for public trust in institutions would enhance the introduction's impact.
The paragraph beginning on line 151 has been expanded to include a reference to this study:
"Respondents who experienced hardship, such as losing a job or going without health care, expressed much less trust in government than those who did not. The result is only correlational. But it raises the possibility of a vicious cycle. Those who experienced hardship were less likely to trust the government, and therefore less likely to comply with Ebola control measures. These people may have put themselves at greater risk of infection. Being infected also meant a higher risk of experiencing further hardship2. The vicious circle model proposed by Blair and colleagues may also apply to health care professionals involved in the pandemic. Struggling physicians may have less confidence in government policies. To them, some policies may seem meaningless. If they become infected by not following these measures, the risk of further difficulties increases. Our study included items related to the number of children in care, self-assessment of socioeconomic status, housing situation and the number of infections or hospitalizations the health worker had suffered as a result of COVID-19. If our data were to support Blair et al.'s vicious cycle model, predictors might include the difficulties identified by such things".
- While the introduction effectively highlights the importance of professionals' trust in government agencies and the need for further research in this area, it would be helpful to outline the specific gaps or knowledge deficits that the current study aims to address. This would provide a clearer rationale for conducting the research and contribute to the significance of the study.
Beginning with line 161, the research gap and contribution are more explicitly stated:
"Studies to date have measured various factors that may contribute to this trust: the use of experts, the promotion of credible sources, the public's belief that professionals are genuinely involved in the decision-making process, and the professionals' own belief that they are involved in the decision-making process. These have increased trust in evidence-based science. This promotes a sense that the health worker is protected. It also encourages the avoidance of biases promoted by the media. It fosters trust in the integrity and ability of institutions to implement effective policies. We are still far from developing a coherent theoretical model that takes all these factors into account. This paper aims to expand the list of factors that may influence health care professionals' trust in public institutions. Professionals’ trust in government agencies has been less studied than public trust".
Literature Review:
- While the introduction references several studies, it lacks a critical analysis and synthesis of existing literature on trust in public health interventions. Provide a more comprehensive review of relevant studies, identifying gaps and inconsistencies, to demonstrate the need for further research.
We have added a paragraph starting at line 165 of the manuscript:
"The literature shows that trust is important for the success of public health interventions during a pandemic. Using different methodologies, the literature identifies different factors that influence trust. The papers focus on public trust in institutions, but not on physicians' trust in institutions in the context of pandemics. We identify two problems. First, little has been written about physicians' trust in institutions in the context of a pandemic. Second, the list of factors already suggested to influence this trust is short. This paper contributes to both issues by identifying (other) factors and by focusing specifically on physicians' trust in government institutions".
The above content is followed by a paragraph starting on line 166 of the manuscript:
"The literature shows that both public trust and health care professionals' trust in public institutions increase the likelihood of effective health care policy implementation. This is even more true during epidemics. Studies to date have measured various factors that may contribute to this trust: the use of experts, the promotion of credible sources, the public's belief that professionals are genuinely involved in the decision-making process, and the professionals' own belief that they are involved in the decision-making process. These have increased trust in evidence-based science. This promotes a sense that the health worker is protected. It also encourages the avoidance of biases promoted by the media. It fosters trust in the integrity and ability of institutions to implement effective policies. We are still far from developing a coherent theoretical model that takes all these factors into account. This paper aims to expand the list of factors that may influence health care professionals' trust in public institutions. Professionals’ trust in government agencies has been less studied than public trust. This paper contributes to the literature gap on factors influencing health care professionals' trust in public institutions implementing health policies. The more we know about the factors that influence health care professionals' trust, the more we can influence these factors in a future epidemic".
- Consider integrating contrasting perspectives or theories to present a balanced view of the topic and enhance the scholarly discussion.
The paragraph starting on line 178 of the manuscript addresses this problem:
"Studies to date have measured various factors that may contribute to this trust: the use of experts, the promotion of credible sources, the public's belief that professionals are genuinely involved in the decision-making process, and the professionals' own belief that they are involved in the decision-making process. These have increased trust in evidence-based science. This promotes a sense that the health worker is protected. It also encourages the avoidance of biases promoted by the media. It fosters trust in the integrity and ability of institutions to implement effective policies. We are still far from developing a coherent theoretical model that takes all these factors into account".
Research Gap and Objectives:
- Clearly state the research gap that this study aims to address within the context of trust in public health interventions. This will help establish the rationale for conducting the research and provide a clear focus for the study.
The paragraph starting on line 165 addresses this issue:
"The literature shows that trust is important for the success of public health interventions during a pandemic. Using different methodologies, the literature identifies different factors that influence trust. The papers focus on public trust in institutions, but not on physicians' trust in institutions in the context of pandemics. We identify two problems. First, little has been written about physicians' trust in institutions in the context of a pandemic. Second, the list of factors already suggested to influence this trust is short".
The paragraph starting on line 178 addresses this issue:
"Studies to date have measured various factors that may contribute to this trust: the use of experts, the promotion of credible sources, the public's belief that professionals are genuinely involved in the decision-making process, and the professionals' own belief that they are involved in the decision-making process. These have increased trust in evidence-based science. This promotes a sense that the health worker is protected. It also encourages the avoidance of biases promoted by the media. It fosters trust in the integrity and ability of institutions to implement effective policies".
- Explicitly state the specific objectives or research questions that will guide the study. This will enhance the clarity of the research goals and align the introduction with the subsequent sections of the manuscript.
The paragraph starting on line 186 addresses this issue:
"This paper aims to expand the list of factors that may influence health care professionals' trust in public institutions. Professionals’ trust in government agencies has been less studied than public trust. This paper contributes to the literature gap on factors influencing health care professionals' trust in public institutions implementing health policies".
Methodological Approach:
- The introduction would benefit from a brief overview of the methodology to be employed in the study. Include information about the research design, sample selection, data collection methods, and analytical techniques to give readers a sense of the empirical approach used in the research.
We included a short paragraph in the introduction about the methodology used in the study, starting on line 198 of the manuscript.
"The aim of the study was to profile physicians willing to raise public health issues with relevant institutions during the COVID-19 pandemic. The online questionnaire consists of questions on socio-demographic and socio-economic status, self-reported diseases, medical specialties, health status, place of work, and involvement in pandemic-related activities. The Likert-type scales measure constructs such as self-efficacy, duty, willingness to work, institutional and governmental reciprocity. A total of 1,285 Romanian physicians participated in the study, and binary logistic regression was used to determine the best predictors for the two categories of responses (higher and lower trust in relevant institutions). A total of 74 items were entered into the regression equation, and significant differences were found between the two categories. The study provides insight into the factors that influence a physician’s willingness to address public health issues during a pandemic".
Language and Style:
- Pay attention to sentence structure and grammar to improve the overall readability and clarity of the text. Some sentences are overly complex and require simplification for better comprehension.
The manuscript has been edited by a native American English speaker. His interventions appear in blue in the manuscript, with changes tracked.
Overall Implications:
- While the introduction touches on the implications of the research, it would be valuable to explicitly state the potential contributions and practical significance of the study. Clearly outline how the findings of the research can inform public health interventions and policy-making processes.
We added a paragraph starting on line 190 of the document:
"The more we know about the factors that influence health care professionals' trust, the more we can influence these factors in a future epidemic. Because the context of an epidemic requires rapid intervention, knowing the factors in advance can pave the way for a rapid response. Changing the factors that influence physicians' trust in government institutions can be done early. If all these trust interventions prove effective, physicians will be more likely to respond favorably to public health interventions when the need arises".
Methods
- Sample Selection: The method mentions that the sample consisted of 1285 Romanian doctors, but it does not provide information on the sampling technique employed. It is essential to explain how the participants were recruited to ensure the representativeness of the sample and generalizability of the findings. Additionally, information on the response rate and any potential biases in participant selection should be addressed.
We added a paragraph starting on line 280 of the manuscript.
"The total number of respondents can be considered a convenience sample, although it is intended to be as close as possible to national representativeness. The questionnaires were distributed via the Internet through the local offices of the professional associations, with the intention of covering as much of the country as possible. The exact response rate is not known. It is difficult to calculate the proportion of respondents among those invited to respond. This is due to the fact that the questionnaire was distributed through professional networks and different branches of professional associations. However, the composition of the sample by region of the country and the urban-rural distribution can be presented. Table 1 shows the distribution of the respondents to the questionnaire by historical regions of the country (Transylvania, Banat, Crișana, etc.) and by type of community (village, small town, medium town, large town). The most important selection bias is the self-selection of the participants. The fact that the questionnaire was completed anonymously attempted to counter this effect. However, self-selection bias cannot be completely eliminated. It is possible that both extremely busy and extremely dissatisfied doctors may have responded in a lower proportion. These biases are a limitation of the study".
We have introduced a new table starting at line 302 of the manuscript.
|
|
Place of medical practice |
Total |
|||
|
village |
small town (10,000-100.000 inhabitants) |
medium-sized city (100,000-500.000 inhabitants) |
big city (over 500.000 inhabitants) |
|
|
|
Transilvania |
22 |
90 |
112 |
74 |
298 |
|
Banat |
4 |
8 |
25 |
32 |
69 |
|
Crișana |
6 |
11 |
29 |
8 |
54 |
|
Moldova |
57 |
41 |
74 |
136 |
308 |
|
Oltenia |
11 |
30 |
39 |
19 |
99 |
|
Muntenia |
28 |
37 |
44 |
256 |
365 |
|
Bucovina |
8 |
22 |
9 |
1 |
40 |
|
Maramureș |
0 |
5 |
12 |
1 |
18 |
|
Dobrogea |
2 |
9 |
8 |
15 |
34 |
|
Total |
138 |
253 |
352 |
542 |
1285 |
- Survey Instrument: The method briefly mentions the inclusion of socio-demographic questions and statements rated on a six-point scale. However, it lacks details regarding the specific items used in the survey. It is crucial to provide a clear description of the survey questions, including their content and intended purpose. Additionally, information on the validity and reliability of the survey instrument should be included to ensure the robustness of the data collected.
We added a paragraph starting on line 225 of the manuscript.
"The questionnaire, administered online, was structured in the following parts. From question 1 to question 18 both socio-demographic items (e.g. gender, age, region in Romania where the doctor works, marital status) and items related to socio-economic status (number of people with whom the doctor lives, number of elderly people in the household, number of young children, number of rooms in the household, self-assessment of economic status on a scale from 1 to 10, whether the doctor is currently still employed, etc.) were included. Question 19 targeted self-reporting of the various diseases the doctor suffers from. Because, in the online questionnaire, the respondent had to tick from a list of multiple options, in the database we unpacked each option into a separate dichotomous variable (e.g. question 19t "Does the medical professional self-report bronchial asthma, including allergic asthma" - answer YES = 1 or NO = 0). Without this decomposition into dichotomous variables, it would have been impossible to process the information resulting from item 19. Further, from item 20 to item 25, questions were asked about the medical professional's specialty and self-assessment of health status in the last two weeks, respectively in the two weeks before the pandemic. Question 26 targeted the specific place of work of the health care professional (e.g. maternity, intensive care, day hospital, etc.). Because two or more of these options could be present at the same time, we split each option into a separate dichotomous variable, similar to item 19. From question 27 to question 36, the healthcare professional was asked whether he/she worked in a quarantined locality, the average number of hours worked per week during the pandemic, whether he/she was hospitalized as a result of a COVID infection, etc. Question 37 referred to the types of activities in which the healthcare professional was involved during the pandemic (e.g. collecting biological samples from suspected COVID-19 patients, performing aerosol-generating procedures, examining a patient with respiratory symptoms, etc.). This question was also broken down into several dichotomous variables, since the same physician performed one or more of these activities. Question 99 asked for a ranking from 1 (most important) to 3 factors that negatively influenced medical practice (e.g. infection control procedures, discharge of non-emergency patients from hospital, etc.). Questions 129 and 130 also asked for a ranking of main sources of information (e.g. direct supervisors, other colleagues in the community, etc.) and main supportive factors when the doctor faced a professional problem (e.g. colleagues at work, professional organizations, etc.). Questions 99, 129 and 130 were in turn broken down into separate variables for each option, where the number corresponding to the hierarchy (1, 2 or 3) was recorded. The remaining questions in the range 38 - 144 were questions answered on 6-point Likert-type scales and referred to various constructs in the literature: self-efficacy, duty, willingness to work, institutional reciprocity and government reciprocity. Due to the lengthy process of constructing the questionnaire, the items were not grouped in the database according to the construct they were part of. However, each item was labelled as such in the database (e.g. Question 49: "I was aware of my role during the pandemic" - Self-efficacy; Question 43: "A healthcare professional who does not have children should work on the front lines during a pandemic" (reversed item) - Duty). After the data collection, due to multiple situations that did not apply to one or more of the physicians, the research team abandoned the aggregation of items into the questionnaires by calculating the internal consistency coefficient and a total per scale. All the data were processed in IBM SPSS Statistics 20.0".
We have expanded the explanations in the paragraph beginning on line 321 of the manuscript.
"In the larger study, the intention was to see if the responses were distributed according to the theoretical models of connection between self-efficacy, willingness to work and duty to care. However, the nature of the data did not allow for linear regression. Most questions had a "not applicable" response option and were not normally distributed. The "not applicable" response choice was introduced in most items, because there were real life situations that did not apply to the respondent. One example is item 48 " When asked to work with COVID-19 patients, I was willing to respond". Since not all physicians were asked to work with COVID-19 patients, there was a "not applicable" response option. Another example is item 53 "I was willing to provide direct patient care even though I did not have access to a K95 mask, although I should have used it.”. Many physicians had access to masks from the very beginning, so they were not confronted with this situation where they had to choose whether to risk their health. We chose logistic binary regression which can process large amounts of data and to select relevant predictors for a given response".
- Data Analysis: While the method mentions the use of logistical binary regression, there is limited explanation of the specific steps and procedures followed in the analysis. It is essential to provide a more detailed description of the regression model, including the independent variables considered, the criteria for variable selection (Backward-Wald computations), and any assumptions or limitations associated with the analysis.
Added a paragraph starting on line 354 of the manuscript:
"The database we worked with contained 143 items. For a first selection, we used Mann-Whitney tests for variables measured by scales and chi-square tests for categorical variables. These showed us the variables for which there was a significant difference between the respondents in the "high confidence" category and those in the "low confidence" category. This resulted in 74 items for which the score was significantly different between the two categories. All were entered into a binomial logistic regression equation. The probability of removal was set at 0.05 and the probability of introduction was also set at 0.05. Due to space limitations, we included the items that were removed in the last five steps of the logistic regression in Table 4".
Added Table 4 starting on line 435 of the manuscript: Items removed in the last five steps of the Bacward-Wald regression
|
Item in the general Questionnaire |
Score |
Sig. |
|
27. Years of medical practice |
.827 |
.363 |
|
45.A young healthcare professional should work on the front line during a pandemic. |
.681 |
.409 |
|
59. My role in the overall response to the pandemic was an important one. |
.569 |
.451 |
|
70.Not showing up for work during the pandemic means abandoning one's team. |
2.467 |
.116 |
|
71.Not showing up for work during a pandemic means abandoning one's patients. |
.319 |
.572 |
- Ethical Considerations: The method section does not mention anything about ethical considerations, such as informed consent and data confidentiality. It is essential to address how ethical guidelines were followed throughout the study, including obtaining informed consent from participants and ensuring the anonymity and confidentiality of their responses.
We have added a paragraph detailing ethical considerations beginning on line 213 of the manuscript.
"The online questionnaire began with an informed consent form that the participant had to agree to. The consent form provided details about the purpose of the study, the research team and the methodology used to collect the data. The form also included the following statement: "None of the information you provide will be linked to you as an individual. [...] There is no risk to the survey other than you may become more aware of the impact the COVID-19 pandemic may have had on your well-being as well as your professional experience during this time". Regarding privacy, the section included the following assurance: "By agreeing to participate in this research, you are not jeopardizing any legal rights. All information from your participation in this research will be collected and stored in accordance with the General Data Protection Regulation (GDPR)." The end of the consent section contained the following statement: "By pressing the CONTINUE button, I agree to participate in this study".
- Limitations: The method section should acknowledge the limitations of the study. For example, potential sources of bias, such as self-reporting bias or non-response bias, should be discussed. Additionally, any constraints related to the study design or data collection process should be acknowledged to provide a comprehensive understanding of the study's scope and potential implications.
As requested, we have added two paragraphs in the Method section, starting on line 290, regarding the limitations of the study
"The most important selection bias is the self-selection of the participants. The fact that the questionnaire was completed anonymously attempted to counter this effect. However, self-selection bias cannot be completely eliminated. It is possible that both extremely busy and extremely dissatisfied doctors may have responded in a lower proportion. These biases are a limitation of the study.
In addition, the fact that the questionnaire was distributed through professional associations and networks may be another source of bias. It is possible that physicians who were better connected to their colleagues in professional associations may have participated more. Physicians who use the Internet less, those who read professional association newsletters less often, and those with fewer connections to colleagues may have been underrepresented".
We have added a paragraph in the Method section starting on line 305 of the manuscript.
"Another shortcoming of the study was the inability to aggregate multiple questionnaire items into constructs. In other words, in most cases we could not measure the same dimension (e.g. self-efficacy) with a total score calculated over several questions. A total score would have required at least an internal consistency calculation and a normality check of the distribution of the total score. Unfortunately, there were too many situations where one or more items did not apply to the participating physician. The total number of the respondents who provided a response for each item in a construct on a Likert-type scale remained very small. The situations in practice turned out to be much more diverse than we had planned".
Results
- The results section lacks crucial details and statistical information, which hinders the understanding and interpretation of the findings. The absence of specific variables and their coefficients or odds ratios in the logistic binary regression model limits the ability to assess the significance and magnitude of the predictors. Providing this information is essential for readers to evaluate the strength and direction of the relationships.
We have introduced the ODDS RATIO column in Table 3.
|
Item in the general Questionnaire |
B coefficient |
Standard Deviation |
Wald |
Odds Ratio |
Sig. |
|
I trusted that I was safe at work during the pandemic. |
.14 |
.05 |
8.78 |
1.15 |
<.01 |
|
The financial bonus we were promised justifies the risk I took. |
.08 |
.04 |
4.59 |
1.09 |
.03 |
|
I have been trained on the use of protective equipment used during the pandemic. |
.16 |
.04 |
19.46 |
1.17 |
<.01 |
|
My colleagues and I share the same values. |
.17 |
.06 |
9.16 |
1.19 |
<.01 |
|
Compared to before the pandemic I enjoy my work just as much. |
.11 |
.05 |
6.09 |
1.12 |
.01 |
|
Constant |
-2.55 |
.32 |
64.10 |
0.08 |
<.01 |
We added a paragraph explaining odds ratios starting on line 381 of the manuscript.
"Given that all odds ratios have values above 1, we conclude that agreement on all predictors listed in Table 3 point in the direction of agreement on the criterion variable question. In other words, agreement for each significant predictor increases the odds that the respective respondent also agrees that he or she can raise public health concerns with the relevant institutions. For example, for the item "I trusted that I was safe at work during the pandemic", the interpretation of the Odds Ratio 1.15 shows that the strength of the association is low. The odds of being among those who are highly confident that they can raise public health issues are 1.15 times higher for those who trusted that they felt safe at work than for those who did not trust that they felt safe at work".
- Additionally, the discussion of pseudo R2 coefficients lacks interpretation and context. Simply stating the values without explaining their meaning and implications diminishes the usefulness of these measures. Moreover, the assertion that the model was more accurate for respondents with lower trust than those with higher trust is mentioned without supporting evidence or further explanation. This discrepancy in model accuracy should be explored and discussed in more depth.
We added a paragraph starting on line 382 and also an extra reference:
"Since R2 cannot be used as a definitive or exact value for the percentage of variance accounted for, the best interpretation of this model is in terms of a modest correspondence between the data and the model resulting from the analysis. The R2 coefficients are indicators of how well the model fits the actual data. The closer the values are to 1.0, the better the model matches the actual data. Values of 0.09 and 0.12 are far from the reference value of 1.0, indicating a modest agreement between the model and the real data16".
We added a paragraph starting on line 389:
"The model indicates the accuracy of the classification. It shows the relative proportions of low and high confidence the participants correctly identified. The model predicts those with low confidence with significantly higher accuracy. This means that low scores on the five predictor variables are more relevant for the low confidence respondents than high scores on the five predictor variables are for the high confidence respondents. From a practical point of view, this may tell us that the tendency to disagree on the items presented as predictors is more relevant for those who do not feel they can raise public health issues with the relevant institutions. Let us consider a hypothetical case: a doctor disagrees with the statement "I trusted that I would be safe at work during the pandemic. He also disagrees with the statements "The financial bonus we were promised justified the risk I took" and "I was trained in the use of the protective equipment used during the pandemic. Finally, the same physician disagrees with the statements "My colleagues and I share the same values" and "Compared to before the pandemic, I enjoy my work just as much". Based on these responses, the model predicts that in 73% of the cases, such a doctor would have little confidence in raising public health issues with the relevant institutions. If another hypothetical physician had responded in agreement with the above statements, the model would have predicted that such a physician would have high confidence in raising public health issues with the relevant institutions only in 53% of cases. In other words, the predictive model is better at identifying those who are distrustful. This would suggest potential strategies for identifying factors that might alter a physician’s propensity to react negatively to such items. According to the predictive model, a significant change in these would lead to a significant change in the confidence. It is important that the model better predicts those with low trust, as changing their perceptions is the goal of any intervention aimed at increasing trust in public institutions".
- Furthermore, the presentation of the predictors and their corresponding statements is insufficient. Merely listing the statements without providing the associated results or discussing their implications limits the reader's ability to fully comprehend the findings. A tabular format displaying the predictors, coefficients, standard deviations, and Wald coefficients would enhance the clarity and accessibility of the results.
We have introduced the ODDS RATIO column in Table 3.
|
Item in the general Questionnaire |
B coefficient |
Standard Deviation |
Wald |
Odds Ratio |
Sig. |
|
I trusted that I was safe at work during the pandemic. |
.14 |
.05 |
8.78 |
1.15 |
<.01 |
|
The financial bonus we were promised justifies the risk I took. |
.08 |
.04 |
4.59 |
1.09 |
.03 |
|
I have been trained on the use of protective equipment used during the pandemic. |
.16 |
.04 |
19.46 |
1.17 |
<.01 |
|
My colleagues and I share the same values. |
.17 |
.06 |
9.16 |
1.19 |
<.01 |
|
Compared to before the pandemic I enjoy my work just as much. |
.11 |
.05 |
6.09 |
1.12 |
.01 |
|
Constant |
-2.55 |
.32 |
64.10 |
0.08 |
<.01 |
Added Table 4 starting on line 435 of the manuscript: Items removed in the last five steps of the Bacward-Wald regression:
|
Item in the general Questionnaire |
Score |
Sig. |
|
27. Years of medical practice |
.827 |
.363 |
|
45.A young healthcare professional should work on the front line during a pandemic. |
.681 |
.409 |
|
59. My role in the overall response to the pandemic was an important one. |
.569 |
.451 |
|
70.Not showing up for work during the pandemic means abandoning one's team. |
2.467 |
.116 |
|
71.Not showing up for work during a pandemic means abandoning one's patients. |
.319 |
.572 |
- To improve the quality of the results section, it is crucial to include specific variables and their coefficients or odds ratios in the logistic binary regression model. Additionally, providing a more comprehensive interpretation of the pseudo R2 coefficients and exploring the discrepancy in model accuracy would enhance the overall analysis. Finally, presenting the predictors and their associated statements in a tabular format with relevant statistical information would improve the clarity and comprehensibility of the findings.
We added a paragraph starting on line 382 and also an extra reference:
Since R2 cannot be used as a definitive or exact value for the percentage of variance accounted for, the best interpretation of this model is in terms of a modest correspondence between the data and the model resulting from the analysis. The R2 coefficients are indicators of how well the model fits the actual data. The closer the values are to 1.0, the better the model matches the actual data. Values of 0.09 and 0.12 are far from the reference value of 1.0, indicating a modest agreement between the model and the real data16.
We added a paragraph starting on line 389:
"The model indicates the accuracy of the classification. It shows the relative proportions of low and high confidence the participants correctly identified. The model predicts those with low confidence with significantly higher accuracy. This means that low scores on the five predictor variables are more relevant for the low confidence respondents than high scores on the five predictor variables are for the high confidence respondents. From a practical point of view, this may tell us that the tendency to disagree on the items presented as predictors is more relevant for those who do not feel they can raise public health issues with the relevant institutions. Let us consider a hypothetical case: a doctor disagrees with the statement "I trusted that I would be safe at work during the pandemic. He also disagrees with the statements "The financial bonus we were promised justified the risk I took" and "I was trained in the use of the protective equipment used during the pandemic. Finally, the same physician disagrees with the statements "My colleagues and I share the same values" and "Compared to before the pandemic, I enjoy my work just as much". Based on these responses, the model predicts that in 73% of the cases, such a doctor would have little confidence in raising public health issues with the relevant institutions. If another hypothetical physician had responded in agreement with the above statements, the model would have predicted that such a physician would have high confidence in raising public health issues with the relevant institutions only in 53% of cases. In other words, the predictive model is better at identifying those who are distrustful. This would suggest potential strategies for identifying factors that might alter a physician’s propensity to react negatively to such items. According to the predictive model, a significant change in these would lead to a significant change in the confidence. It is important that the model better predicts those with low trust, as changing their perceptions is the goal of any intervention aimed at increasing trust in public institutions".
We have introduced the ODDS RATIO column in Table 3.
|
Item in the general Questionnaire |
B coefficient |
Standard Deviation |
Wald |
Odds Ratio |
Sig. |
|
I trusted that I was safe at work during the pandemic. |
.14 |
.05 |
8.78 |
1.15 |
<.01 |
|
The financial bonus we were promised justifies the risk I took. |
.08 |
.04 |
4.59 |
1.09 |
.03 |
|
I have been trained on the use of protective equipment used during the pandemic. |
.16 |
.04 |
19.46 |
1.17 |
<.01 |
|
My colleagues and I share the same values. |
.17 |
.06 |
9.16 |
1.19 |
<.01 |
|
Compared to before the pandemic I enjoy my work just as much. |
.11 |
.05 |
6.09 |
1.12 |
.01 |
|
Constant |
-2.55 |
.32 |
64.10 |
0.08 |
<.01 |
Added Table 4 starting on line 435 of the manuscript: Items removed in the last five steps of the Bacward-Wald regression
|
Item in the general Questionnaire |
Score |
Sig. |
|
27. Years of medical practice |
.827 |
.363 |
|
45.A young healthcare professional should work on the front line during a pandemic. |
.681 |
.409 |
|
59. My role in the overall response to the pandemic was an important one. |
.569 |
.451 |
|
70.Not showing up for work during the pandemic means abandoning one's team. |
2.467 |
.116 |
|
71.Not showing up for work during a pandemic means abandoning one's patients. |
.319 |
.572 |
Discussion
- The interpretation of the Cox and Snell pseudo R2 coefficients could be further discussed. While the modest fit between the model and the actual data is acknowledged, the implications of this modest fit on the generalizability and reliability of the findings should be addressed. Additionally, the limitations of using binary logistic regression in capturing complex relationships and potential confounding factors should be acknowledged.
We have expanded the paragraph starting on line 608 of the manuscript:
"Despite the large number of respondents, there are several limitations to this study. An obvious limitation of the study is that the Cox and Snell pseudo R2 and Nagelkerke R2 coefficient values were 0.09 and 0.12, respectively. This means that our results can mainly be used as indicators of trends rather than clear differences between categories of people. These R2 values mean that the model explains at most 12% of the variance of the criterion variable. This proportion is low and may indicate that other variables can be found to explain more of the variance of the criterion variable. The modest fit of the model to the actual data suggests a cautious interpretation of the results. The study captured a relatively weak relationship between the predictors and the criterion. This relationship may serve more as a suggestion for further research. Furthermore, the validity of the approach may be affected by possible confounding variables. Because a correlational relationship does not imply a causal relationship, other variables may be found to be related to both the predictor variables and the criterion variable. Such relationships would require, for example, structural equations that capture moderation and mediation phenomena. Therefore, the conclusions of this paper should be generalized with caution. The findings in this paper need to be supported by other research that uncovers relationships between trust and the variables we have mentioned. In addition to this limitation, it should be noted that the criterion variable in this study was measured by a single item. The responses to this question were divided into two categories by the median test, resulting in a dichotomous variable. Further research may identify other relevant predictors that explain more of the variance in the criterion variable. At the same time, a more precise measure of the level of confidence physicians have in public institutions to address public health issues during epidemics could be obtained. This more precise measurement can be achieved by using multiple items that refer to the same construct (e.g., physicians' trust in public institutions) ".
- The discussion on the relationship between feeling safe at work and trust in relevant institutions could benefit from a more nuanced analysis. While it is suggested that feeling safe at work may be a result of personal compliance and effective institutional policies, it would be valuable to explore the potential role of external factors such as resource allocation, communication strategies, and transparency in fostering trust.
We have added a paragraph starting on line 461 of the manuscript:
"Other external factors may play a role in the association we found. First, we hypothesize that the allocation of resources during a pandemic plays an important role in ensuring the safety of health care workers. Therefore, the relationship between trust in public institutions and feeling safe at work could also be explained by this variable. We could assume that effective resource allocation leads to increased safety of health care workers during an epidemic. This increased safety could, in turn, be associated with increased trust in the government institutions that provided this resource allocation. Second, the communication strategies of public institutions could directly influence trust in institutions. But this influence could also be achieved through a sense of security. Especially in a pandemic context such as COVID-19, communicating clear measures about the new virus could make health workers feel safer. A sense of security could therefore increase health workers' confidence that institutions are open to their feedback on relevant public health issues. Finally, the perceived transparency of institutions could affect trust both directly and through a sense of security. We hypothesize that if healthcare professionals perceive a lack of transparency, this could lead them to think of dangers that are not readily disclosed by the authorities in order to avoid panic. Thus, a lack of transparency would lead to greater distrust of institutions by healthcare workers through job insecurity".
- The interpretation of the relationship between financial bonuses and trust in government institutions could be further elaborated. The authors mention that the financial bonus may have a paradoxical effect on trust, but it would be beneficial to delve into the underlying reasons for this effect. Additionally, the potential influence of other factors, such as the perceived fairness of the bonus distribution and the overall support provided to healthcare workers, should be considered.
We have added a paragraph starting on line 505 of the manuscript.
"It seems plausible that physicians who are distrustful of public institutions may feel that the financial reward they receive is too small compared to the effort they put in during the pandemic. The act of receiving a (possibly inadequate) financial reward for a very large extra effort has two sides. Some physicians may perceive that governmental institutions recognize the extra effort. Other physicians, on the other hand, may perceive that they have received a sum of money instead of real recognition for the sacrifice they have made. In some cases, therefore, the lack of financial compensation may prove more useful than financial compensation that is disproportionately small in relation to the effort invested. When there is no form of financial reward, it leaves room for society to recognize a possible form of heroism. If there is some form of financial reward, it is more difficult for those on the sidelines to interpret the act as heroism. This could be a hypothetical mechanism for the paradoxical effect we have observed".
- The discussion on the importance of shared values among physicians and its impact on trust in relevant institutions is intriguing. However, the practical implications and feasibility of organizing specific events to promote a sense of belonging and shared values need further exploration. The authors could provide more details on how such events could be structured and the potential challenges that may arise in implementing them.
We have added a paragraph starting on line 571 of the manuscript.
"The corporate environment has brought the idea of team building to Romania. It is well known that the purpose of such events is to strengthen the feeling of belonging to a team. It also aims to improve the quality of communication and the ability to work as a team. On different sides of imaginary barriers, doctors can forget they share the same values. And, in a crisis situation, such as the COVID-19 pandemic, this brings to light possible negative images that doctors may have of public institutions. Case studies play a special role in various types of training. Learning and team-building situations can be designed in which teams of five people work on a case study of the response to an epidemic. Team members are recruited both from doctors in hospitals or clinics and from specialists in government institutions. The case can be thought of in such a way that the whole team is in the position of a crisis cell. This crisis cell has to coordinate communication with the public, communication with hospitals, implementation of public health measures, budget optimization and so on. The team members are thus confronted with the different types of challenges that arise in such situations. They can realize that the whole team is trying to maximize the good for society as a whole. Each team member would have the opportunity to realize that he or she has similar or identical values to their colleagues. The urgency of a situation is not a prerequisite for the abandonment of these values. On the contrary, teams of people seek the best way to achieve an optimal outcome".
- The limitations of the study are briefly mentioned but could be expanded upon. The potential sources of bias, such as self-reporting and the reliance on a specific sample of Romanian doctors, should be acknowledged. Additionally, the generalizability of the findings to other contexts and healthcare systems should be addressed.
We added several paragraphs starting on line 608 of the manuscript.
"Despite the large number of respondents, there are several limitations to this study. An obvious limitation of the study is that the Cox and Snell pseudo R2 and Nagelkerke R2 coefficient values were 0.09 and 0.12, respectively. This means that our results can mainly be used as indicators of trends rather than clear differences between categories of people. These R2 values mean that the model explains at most 12% of the variance of the criterion variable. This proportion is low and may indicate that other variables can be found to explain more of the variance of the criterion variable. The modest fit of the model to the actual data suggests a cautious interpretation of the results. The study captured a relatively weak relationship between the predictors and the criterion. This relationship may serve more as a suggestion for further research. Furthermore, the validity of the approach may be affected by possible confounding variables. Because a correlational relationship does not imply a causal relationship, other variables may be found to be related to both the predictor variables and the criterion variable. Such relationships would require, for example, structural equations that capture moderation and mediation phenomena. Therefore, the conclusions of this paper should be generalized with caution. The findings in this paper need to be supported by other research that uncovers relationships between trust and the variables we have mentioned. In addition to this limitation, it should be noted that the criterion variable in this study was measured by a single item. The responses to this question were divided into two categories by the median test, resulting in a dichotomous variable. Further research may identify other relevant predictors that explain more of the variance in the criterion variable. At the same time, a more precise measure of the level of confidence physicians have in public institutions to address public health issues during epidemics could be obtained. This more precise measurement can be achieved by using multiple items that refer to the same construct (e.g., physicians' trust in public institutions).
Another limitation of the survey is the "not applicable" option in the answer to most questions. As explained in the method section, this option did not illustrate a neutral response. It was introduced because, from a practical point of view, some situations did not apply to the respondent. For example, to the question "My family was prepared to manage without me while I worked during the pandemic", some doctors answered "not applicable", as they were away from their families anyway. To the question "My employer and I share the same values", some doctors answered "not applicable" because they were self-employed. From this, it resulted that the statistical processing program included only 902 cases in the analysis, i.e. 70.2% of all the respondents. The number of missing cases was 383. Therefore, the results apply to the total of 902 cases that were included in the analysis and not to the total of 1,285. From this point of view, the statistical power and representativeness of the sample may have been diminished.
As we noted in the Method section, selection bias in the self-selection of the participants remains the most significant limitation of this study, despite the use of anonymous surveys to counteract its effects. Busy or dissatisfied physicians may have been underrepresented in the responses. In addition, the distribution of surveys through professional associations and networks may have introduced a bias toward physicians with better connections within these networks. Internet use, frequency of newsletter reading, and professional connections may have influenced participation rates. Another limitation was the inability to aggregate questionnaire items into constructs such as self-efficacy. Due to numerous instances where survey questions did not apply to the physician, the total number of respondents per construct was small.
The study was conducted on a Romanian sample. There are two difficulties regarding the possible generalization of the results. First, there is the rather low power of the model to make predictions, even in the Romanian physician population. The R-squared coefficients are low. There are possible confounding variables, as we already showed in the Discussion section. Moreover, the representativeness of the sample for the whole population of Romanian doctors is good but not perfect. Secondly, since we are dealing with a group of Romanian doctors, we cannot claim generalizations to doctors in other countries. Different cultures may perceive the factors influencing medical staffs’ trust in government institutions quite differently. Only by finding the same relationships in other cultures can we hypothesize about Western doctors' trust in government institutions. Other studies might use standardized questionnaires. These questionnaires can be adapted in other cultures to test the same set of hypotheses. Only in this way can we build general theoretical models for Western physicians' trust in government institutions in a pandemic situation".
Conclusion
- The conclusion does not adequately discuss the limitations of the study. It is important to acknowledge any potential biases, limitations in data collection or analysis, and the generalizability of the findings. This would provide a more balanced interpretation of the results and acknowledge the study's shortcomings.
We added two paragraphs in the Conclusion section, starting on line 702:
"As shown above, the use of logistic regression had both advantages and disadvantages. The logistic regression equation allowed for the processing of a large number of variables and avoided false positives through the nonparametric approach. However, the associations identified were modest and could not explain any of the possible causal relationships. An alternative approach would be to construct questionnaires with general statements about trust in public institutions (e.g., "I believe that public institutions will tell the truth when implementing health care policies during a pandemic"). These general statements would no longer require a "not applicable" option. Several such items could be aggregated into constructs with theoretical validity and internal consistency. A normal distribution of the scores would allow either linear regressions or structural equations. The latter could indicate a possible theoretical explanatory model much better than logistic regression.
This study has other limitations that must be considered. The most important limitation is the self-selection bias of the participants. Another limitation is that the surveys were distributed through professional associations and networks, resulting in a bias toward better-connected physicians. The study also has an inability to aggregate multiple questionnaire items into constructs, resulting in single-item measures. The pseudo-R2 coefficient values are low, indicating that the model can explain at best 12% of the variance in the criterion variable. The study may be affected by confounding the variables. In addition, the criterion variable was measured by a single item and the study had a high number of missing cases. As the study was conducted in Romania, the generalizability of the results to other cultures or countries may be limited. Therefore, this paper emphasizes the need for further research in order to establish relationships between trust and the variables mentioned. This can be done by using standardized questionnaires that can be adapted to different cultures. General theoretical models of physicians' trust in government institutions in a pandemic situation could be developed".
- The conclusion does not clearly establish causality between the identified predictors and the ability of physicians to raise public health issues. While the logistic regression model identifies associations, it is crucial to emphasize that correlation does not imply causation. The conclusion should exercise caution in making definitive claims about the factors influencing physicians' trust and ability to raise concerns.
We introduced a paragraph starting on line 679.
"It is important to note that a predictive model can only identify possible relationships between variables. Obviously, no correlation system can claim to identify cause-and-effect relationships. As with any study of this nature, the results must be interpreted with caution. Our findings can only suggest new directions for research and possible interventions. These suggestions may be confirmed by other findings with more controlled research designs".
- The conclusion lacks a critical reflection on the methodology used. It would be valuable to discuss potential alternative approaches or limitations of using binary logistic regression as the primary analytical method. Additionally, suggestions for future research directions or methodological improvements could be included to encourage further investigation.
We added a paragraph starting on line 702:
"As shown above, the use of logistic regression had both advantages and disadvantages. The logistic regression equation allowed for the processing of a large number of variables and avoided false positives through the nonparametric approach. However, the associations identified were modest and could not explain any of the possible causal relationships. An alternative approach would be to construct questionnaires with general statements about trust in public institutions (e.g., "I believe that public institutions will tell the truth when implementing health care policies during a pandemic"). These general statements would no longer require a "not applicable" option. Several such items could be aggregated into constructs with theoretical validity and internal consistency. A normal distribution of the scores would allow either linear regressions or structural equations. The latter could indicate a possible theoretical explanatory model much better than logistic regression".
- The conclusion does not address potential ethical considerations. It is important to discuss the ethical implications of the findings, such as the potential impact on patient care, the duty of physicians to advocate for public health, and the responsibility of institutions to foster an environment conducive to raising concerns.
We have added a final paragraph starting at line 748 of the manuscript.
"The results of this study can be discussed from an ethical perspective. Institutions have a responsibility to create an environment that encourages physician feedback. In return, medical staff implement public health policies and measures. Only through this "contract" is the patient truly protected. One of the physician's duties is to advocate for his or her patients. The confidence to raise public health issues with the relevant institutions is essential for the physician to do his or her job. If further studies confirm the importance of the factors we have identified, patient care may be indirectly improved by influencing these factors".
- The conclusion could benefit from a more nuanced discussion of the implications for policy and practice. Merely suggesting that policymakers focus on specific factors without delving into the challenges and practicalities of implementing these recommendations may limit the usefulness of the findings.
We added a paragraph starting on line 728:
"Not all of the factors we identified can be directly addressed by government policy. If the results of our study prove to be valid, some direct steps can be taken to increase physician confidence. Our findings suggest that as long as physicians feel that the financial reward justifies the risk, physician confidence will increase. A challenge in this regard would be to identify the point at which a financial reward "justifies" the perceived risk to the physician, given budgetary constraints. Second, training in the use of safeguards can be promoted by the central authorities. This training can be provided on a regular basis as part of a program to ensure responsiveness to a pandemic. Government institutions can ensure that such programs are conducted either in hospitals or in professional schools. However, government institutions cannot guarantee the quality of the training programs unless the trainers come from the center. The latter is more difficult to implement in practice. Instead, the subjective feeling of being safe at work can depend on many factors. Not all of these factors can be controlled by government institutions or the institution where the doctor works. The most difficult to influence is the sense of shared values between the physician and colleagues, as well as the intrinsic emotional rewards of practicing medicine. Programs can be created, to cultivate a sense of belonging to a shared value system, but finding ways to increase intrinsic emotional rewards seems difficult. Many variables related to the physician's personality and value system come into play. Obviously, part of the physician's confidence that he or she can raise public health issues with the appropriate institutions is also influenced by deeply personal variables".
Round 2
Reviewer 2 Report
The authors have met most of my concerns. I recommend the publication of this manuscript in its current form.
Thanks